# FROM ACCELERATION TO SATURATION: SCALING BEHAVIOR OF BOOTSTRAPPED LANGUAGE MODEL PRETRAINING

## ABSTRACT

Bootstrapped pretraining, i.e., the reuse of a pretrained base model for further pretraining, such as continual pretraining or model growth, is promising at reducing the cost of training language models from scratch. However, its effectiveness remains unclear, especially when applied to overtrained base models. In this work, we empirically study the scaling behavior of bootstrapped pretraining and find that its scaling efficiency diminishes in a predictable manner: The scaling exponent with respect to second-stage pretraining tokens decreases logarithmically with the number of tokens used to pretrain the base model. The joint dependence on first- and second-stage tokens is accurately modeled by a simple scaling law. Such saturation effect reveals a fundamental trade-off in multi-stage pretraining strategies: the more extensively a model is pretrained, the less additional benefit bootstrapping provides. Our findings provide practical insights for efficient language model training and raise important considerations for the reuse of overtrained models.

## 1 INTRODUCTION

Large language models (LLMs) have recently shown astounding performance in various natural language processing tasks, reaching human-level capabilities in some cases (Achiam et al., 2023; Anthropic, 2024; Gemini Team Google, 2023). However, training/pretraining these models from scratch is computationally expensive and time-consuming, requiring weeks to months even with powerful GPU clusters. To address this challenge *within the pretraining stage itself*, researchers have explored strategies for bootstrapping/reusing existing pretrained (base) models for various purposes. These include strategies to learn with new pretraining data, or to increase the model size, without starting from scratch. We refer such strategies that pretrain a model in multiple stages collectively as *bootstrapped pretraining*.

More specifically, we are interested in bootstrapped pretraining strategies that (1) improve domain-specific performance through *continual pretraining* (CPT) of the base model (see, e.g., Ibrahim et al. (2024)); or (2) scale model capacity via *model growth* techniques, which increase the model size reusing base model parameters to speed up training (Chen et al., 2015). These strategies have been shown to be promising at accelerating performance improvement of LLMs while reducing the computational cost compared to pretraining from scratch.

On the other hand, when a base model is overtrained, intuitively, its highly optimized parameters make effective exploration after capacity enlargement for model growth (or exploration with new data distribution for CPT) difficult, causing the second stage to learn slower. This is known as the loss of plasticity for neural networks over changing tasks (Berariu et al., 2021; Ash & Adams, 2020; Sodhani et al., 2020). Hence, it is unclear if the second stage of pretraining can be effectively applied to language models that are overtrained, or whether the scaling behavior remains consistent across different training stages and model sizes.

In this paper, we seek to understand the scaling behavior of bootstrapped pretraining methods by running a multitude of controlled language modeling experiments. We find that, from the performance (cross-entropy loss) perspective, bootstrapping overtrained base models leads to *saturation* in scaling during the second-stage pretraining. Specifically, we show that this effect is *quantifiable* via scaling laws: **The scaling exponent (with respect to the number of training tokens in the second stage)**

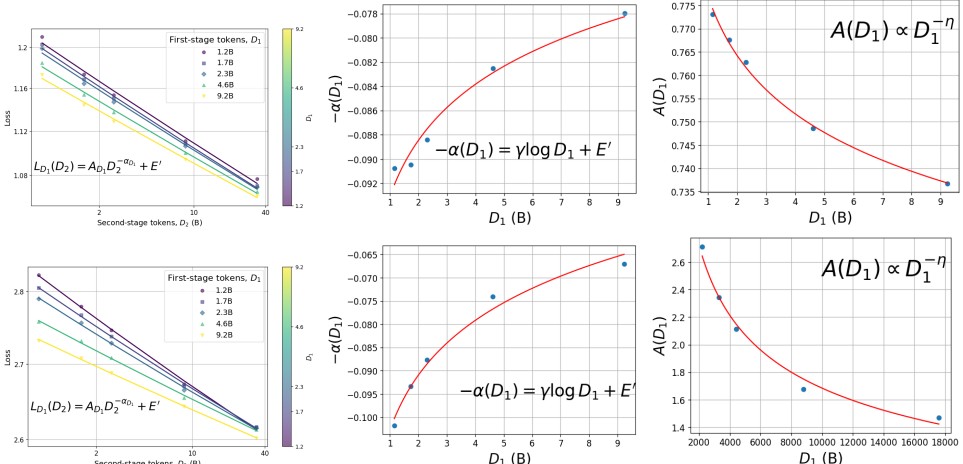

Figure 1: **Bootstrapped pretraining with overtrained base models leads to saturation in scaling behavior. Left: $D_2$ has power-law scaling.** We show scaling behavior of second-stage training tokens ($D_2$) for different values of first-stage tokens ($D_1$), trained on a 0.1B base model. **Middle: Interaction term explains decreasing exponents.** The fitted exponents in the left plots are used to fit Equation 3 as a function of $D_1$, and are shown to agree well with the functional form. **Right: Scaling factor has power-law scaling w.r.t. $D_1$. Top: Continual pretraining (CPT) on code data. Bottom: Model growth from 0.1B to 0.2B by stacking.**

**decreases as the base model is trained for longer periods of time. The decrease is proportional to the logarithm of the number of training tokens invested to the first stage.** See Figure 1. Our work bears practical implications as even relatively tiny models are overtrained to trillions of tokens nowadays (Zhang et al., 2024b; Dubey et al., 2024), and bootstrapped pretraining is widely used in practice.

**Summary of contributions.** More technically, our key contributions are as follows:

- We conduct extensive experiments on language models of various sizes and pretraining tokens/datasets (with over 450 runs), to study the scaling behaviors of bootstrapped pretraining methods, including continual pretraining and model growth.

- We find that for a wide range of configuration of these methods, the following empirical scaling relation holds:

$$L(D_1, D_2) = A D_1^{-\alpha_1} D_2^{-\alpha_2 + \alpha_3 \log D_1} + E \qquad (1)$$

where $L$ is the validation loss after the second stage, $D_1$ and $D_2$ are the number of training tokens in the first and second stages, respectively, and $A, \alpha_1, \alpha_2, \alpha_3, E$ are *positive* constants. We denote the term $\alpha_3 \log D_1$ as the *interaction term*. The scaling exponent with respect to $D_2$ then quantifies the saturation effect of bootstrapped pretraining. We also formulate other plausible functional forms to make comparisons, showing that the above formula still best fits the empirical observation.

- We provide a mechanistic explanation of the saturation effect through analyzing the gradient norms of overtrained models. Then, we demonstrate how the scaling laws can provide guidance on when to train from scratch instead of bootstrapping to mitigate the saturation effect.

- We further analyze the scaling behavior with respect to model size, and show that the scaling laws can be extended to jointly incorporate model size together with dataset sizes, see Equation 10.

- Limitations of our scaling laws for models trained with an extremely large number of tokens are also discussed.

Notations used throughout the paper are summarized in Appendix A.

## 1.1 RELATED WORK

**Continual pretraining.** CPT has been widely used to adapt existing LLMs to specific domains (Sun et al., 2020; Jang et al., 2022a;b), including code (Yadav et al., 2023; Zan et al., 2022; Guo et al., 2024) and mathematics (Gong et al., 2022; Shao et al., 2024) domains, to be studied in this work. Systematic studies at scale are relatively fewer, and our methodology mainly follows Ibrahim et al. (2024).

**Model growth.** While Chen et al. (2015) was the first model growth work in the deep learning era, the idea of growing neural networks from smaller ones can be traced to the 90s (Fahlman & Lebiere, 1989; Fahlman, 1990). More recent language model-related model growth methods include Gong et al. (2019); Chen et al. (2021); Wang et al. (2023a;b); Shen et al. (2022); Evci et al. (2022); Yao et al. (2024), which were subsequently systematically analyzed in Du et al. (2024), motivating our choice of model growth methods.

**Power-law ansatz.** We assume that the validation loss $L$ follows the widely observed *power-law scaling* with respect to a single variable of interest, such as the number of training tokens or the model size (Hestness et al., 2017; 2019; Henighan et al., 2020):

$$L = AX^{-\alpha} + E, \tag{2}$$

where $X$ is the variable of interest, $\alpha$ is the scaling exponent, $A$ is the scaling factor, and $E$ is the irreducible loss due to the inherent entropy of the data distribution. [1]

**Scaling laws.** We focus on modeling only the *final* loss, which has a power-law like behavior as discussed above, a generic phenomenon not only occuring in neural networks, but is also observed in other natural as well as man-made phenomena (Clauset et al., 2009). We shall additionally note that recent scaling law studies attempted to fit the whole loss curve albeit with more sophisticated functional forms Tissue et al. (2024); Wang et al. (2025); Qiu et al. (2025).

Similar to CPT, the reduced capability of transfer learning of code data from models pretrained on natural language data was observed in Hernandez et al. (2021), where the authors denoted as ossification. A similar phenomenon was also observed in the pretraining-fine-tuning pipeline (Springer et al., 2025). These studies however did not quantify the saturated scaling behavior. Previous fine-tuning scaling studies (Mikami et al., 2022; Zhang et al., 2024a; Bethune et al., 2025) did not observe the subtler saturation effects as well, which is perhaps due to the smaller scale (in terms of training tokens) of their experiments.

## 2 EXPERIMENTS

### 2.1 SETUP

**Model configuration.** We consider decoder-only transformers (Vaswani et al., 2017) pretrained with an autoregressive language modeling objective. Our architecture is LLaMA-like (Touvron et al., 2023), incorporating refinements such as SwiGLU activation functions (Shazeer, 2020) and rotary position embeddings (Su et al., 2024). We use the LLaMA tokenizer with a vocabulary size of 32,000. We consider a suite of model sizes up to 1.1B in our experiments. Table 2 of Appendix B contains the model configuration used in this paper.

**Training configuration.** All experiments are conducted using the Megatron-LM library (Shoeybi et al., 2019). We train models in mixed precision (bfloat16) using the AdamW optimizer (Loshchilov et al., 2017), with maximum learning rates tuned separately for each model size. The learning rate follows the *warmup-stable-decay* (WSD) schedule (Bi et al., 2024; Hu et al., 2024), with the final learning rate decayed to one tenth of the peak value. This schedule enables long training runs and facilitates checkpoint reuse for emulating shorter effective training budgets, reducing computational cost. In Appendix C, we also provide an ablation study with the cosine learning rate schedule to

---

[1] Strictly speaking, we assume the form $L = \frac{A}{(X+1)^\alpha} + E$, which ensures finiteness at $X \to 0$. However, since in practice $X \gg 1$ (typically $10^6$ or more), we approximate it as $L \approx AX^{-\alpha} + E$ for notational simplicity. Moreover, we use the same symbols $A$, $\alpha$, and $E$ to denote the scaling factor, exponent, and additive constant, respectively, for notational conveniences when there is no ambiguity, although they may differ across different variables of interest.

show that they achieve similar performance. We train the models for a number of tokens to at least roughly 200 times the model size in tokens in total (a few hundred B tokens overall). Tables 3 and 4 in Appendix summarize the main training configurations. See also Appendix B for further experimental details and Appendix C for training details and evaluation on downstream tasks.

**Two stages of pretraining.** In the context of bootstrapped pretraining, we are primarily interested in how the loss behaves with respect to the number of training tokens used in the two stages of pretraining:

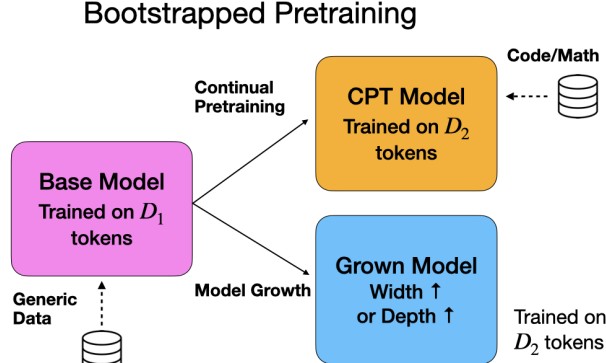

- $D_1$: the number of tokens used in the **first-stage** pretraining (base model).

- $D_2$: the number of tokens used in the **second-stage** pretraining (CPT or model growth).

**Base model.** For the first-stage pretraining, we use the CommonCrawl portion of the Slimpajama-DC dataset (Shen et al., 2023), containing 368B tokens in total. The models trained in this stage serve as base checkpoints for the second-stage bootstrapped pretraining experiments.

Figure 2: **Illustration of bootstrapped pretraining in consideration.** Bootstrapped pretraining consists of two stages: (1) first-stage pretraining of a base model for $D_1$ tokens on internet/generic data; (2) bootstrapping/second-stage pretraining via continual pretraining or model growth for $D_2$ tokens. Section 3 and 3.2 study and develop scaling laws as a function of these two variables (and additionally model size, $N$ in Section 4) to predict the final loss after the second stage.

**Continual Pretraining.** CPT refers to further training a pretrained model on a large dataset, typically billions of tokens, with the goal of improving performance on a new domain. We distinguish CPT from *instruction tuning*, which typically trains with smaller datasets (millions of tokens or fewer), and is thus harder to analyze the scaling behavior. For our experiments, we perform CPT on the base model on two domain-specific datasets: **Code corpus:** Stack/StarCoder (Li et al.); and **mathematics corpus:** OpenWebMath (Paster et al., 2023). Unless otherwise noted, we adopt the same optimizer and learning rate configurations as the first-stage pretraining.

**Model growth.** Model growth refers to increasing model capacity by adding new layers and/or expanding hidden dimensions, thereby increasing the number of trainable parameters. This strategy aims to leverage the representations learned during first-stage training, allowing the larger model to accelerate learning in the second stage. We investigate two model growth techniques shown to be the most effective (in terms of adding new parameters in the width and depth directions) in prior work (Du et al., 2024) (see also Appendix C.5 for more details):

- **Width expansion (exp):** Add new neurons to each layer while using *function-preserving initialization* (FPI), ensuring that the expanded model initially reproduces the behavior of the smaller model (Chen et al., 2021; Du et al., 2024).

- **Depth-wise stacking (stk):** Add new layers to the top of the transformer by copying existing layer weights, thereby extending model depth (Chen et al., 2021; Du et al., 2024) .

After performing this procedure, the larger models are trained on the same dataset as the base model. The growth factor is defined as the ratio of the number of non-embedding parameters in the grown model to that in the base model. For both bootstrapped pretraining scenarios in consideration, the validation loss is evaluated on a held-out set from the second-stage dataset. We illustrate the whole framework in Figure 2.

## 2.2 Experimental Observations

We here provide experimental observations that motivate our formulation of the scaling law. In Figure 1, we show results of training a base model of size 0.1B, and performing CPT on code data, and stacking with a growth factor of 2 on a $5 \times 5$ grid of $D_1, D_2$ [2]. In the left panels, we plot the second-stage validation loss after the second stage as a function of $D_2$ for different values of $D_1$ (left). The following observation is made:

**Observation 1:** The loss is observed to follow a power law with respect to the number of tokens $D_2$, i.e., Equation 2.

Note that this is consistent with classical neural scaling laws and reflects the expectation that, when starting from a fixed initialization, additional tokens improve performance predictably.

Furthermore, we observe that the fitted scaling exponent of Equation 2 decreases as $D_1$ increases, with the model growth method more pronouncedly so. To quantify this trend, we express (the minus of) the scaling exponent as a function of $D_1$, denoted by $-\alpha(D_1)$. A scatter plot of $-\alpha(D_1)$ reveals a clear logarithmic dependence:

$$-\alpha(D_1) = \gamma \log D_1 + E'. \tag{3}$$

This relationship fits the empirical data well, as shown in the same Figure. Substituting Equation 3 into Equation 2, we arrive at a term of the form $D_2^{-E' + \gamma \log D_1}$. Additionally in the same Figure, we show that the multiplicative dependence, $A \propto D_1^{-\alpha_1}$, also holds well. These observations lead to the following:

**Observation 2:** The dependencies of the second-stage loss jointly on $D_1$ and $D_2$ can be captured by a multiplicative scaling law with an interaction term, i.e., Equation 1.

**Interpreting the scaling law.** Several quantitative and qualitative insights can be made from the scaling law. Fixing $D_1$, the effective scaling factor for $D_2$ becomes $A D_1^{-\alpha_1}$. As $D_1$ increases, this factor decreases, resulting in a lower initial loss at the start of second-stage pretraining, agreeing with the conventional wisdom that better-pretrained base models (larger $D_1$) provide stronger initialization for second-stage pretraining.

However, also at fixed $D_1$, the effective scaling exponent with respect to $D_2$ is given by $\alpha_2 - \alpha_3 \log D_1$. This implies that as the base model becomes more overtrained (i.e., larger $D_1$), the improvement in loss from additional second-stage tokens becomes increasingly marginal. In other words, the returns from increasing $D_2$ diminish, manifesting as saturation effects at higher $D_1$.

## 3 Scaling Laws for Bootstrapped Pretraining

### 3.1 Functional Forms for the Scaling Laws

The observations from the previous Section regarding the functional form of the scaling law are purely empirical, which naturally raises the question of whether other functional forms might also fit the observed data. To explore these possibilities, we first derive functional forms that satisfy certain reasonable assumptions, and later fit and compare them empirically with our experimental results.

To recite, our goal is to find a functional form $L(D_1, D_2)$ that captures how the second-stage loss depends jointly on both stages. To derive a principled formulation, we impose two natural constraints:

**Condition 1:** For a fixed base model trained on $D_1$ tokens, the second-stage loss should follow a power law with respect to the number of tokens $D_2$:

$$L(D_1, D_2) = L_{D_1}(D_2) = A_{D_1} D_2^{-\alpha_{D_1}} + E_{D_1}. \tag{4}$$

This is consistent with classical neural scaling laws and Observation 1. It reflects the expectation that, when starting from a fixed initialization, additional tokens improve performance predictably.

**Condition 2:** For any fixed value of $D_2$, the loss should exhibit power-law behavior with respect to the number of first-stage tokens $D_1$:

$$L(D_1, D_2) = L_{D_2}(D_1) = A_{D_2} D_1^{-\alpha_{D_2}} + E_{D_2} \tag{5}$$

---

[2]We double the number of non-embedding layers for stacking

This is consistent with the power-law ansatz and captures the intuition that a better-trained base model (i.e., larger $D_1$) should result in lower loss. Moreover, this condition implies that as $D_2 \to 0$, the loss should continuously approach that of the base model: $\lim_{D_2 \to 0} L(D_1, D_2) = AD_1^{-\alpha} + E$, which is a natural requirement for: [3]

- **CPT:** The second stage's initial loss should begin from the base model's (evaluated on second-stage dataset).
- **Function-preserving model growth:** When model capacity is expanded but initialized carefully, the initial loss should remain close to the base model's loss.

Together, these conditions lead to the following candidate formulations that jointly satisfy both requirements:

$$\textbf{Multiplicative:} \quad L(D_1, D_2) = AD_1^{-\alpha_1} D_2^{-\alpha_2 + \alpha_3 \log D_1} + E. \tag{6}$$

$$\textbf{Additive:} \quad L(D_1, D_2) = AD_1^{-\alpha_1} + FD_2^{-\alpha_2} + E. \tag{7}$$

$$\textbf{Hybrid:} \quad L(D_1, D_2) = \left(AD_1^{-\alpha_1} + F\right)D_2^{-\alpha_2} + E. \tag{8}$$

Note that since $D_1^{-\alpha_1} D_2^{-\alpha_2 + \alpha_3 \log D_1} = D_1^{-\alpha_2 + \alpha_3 \log D_2} D_2^{-\alpha_2}$, the multiplicative form with an interaction term satisfies our conditions, while other forms do not. We further note that multiplicative scaling laws have been observed in Mikami et al. (2022); Zhang et al. (2024a) but without the interaction term. Liew et al. (2025) empirically showed that sparse upcycling (training sparse mixture-of-experts (MoE) models reusing existing dense models) follows a scaling law similar to Equation 6 but under different motivation and conditions. Our work directly extends their findings to other pretraining paradigms. Equation 8 has also been studied in Barnett (2024).

### 3.2 FITTING SCALING LAWS

To determine the functional form described in the previous Section that best describes the scaling behavior of bootstrapped pretraining methods, we consider a variety of datasets and methods beyond those described in the previous Section, using the same 0.1B base model.

**CPT.** We consider both code and mathematics datasets and perform CPT on them as in the previous Section. Additionally, we consider the following variants of CPT that are commonly used in practice (Ibrahim et al., 2024) and test them on the code dataset:

- **CPT with replay (rep):** A portion of the first-stage data is mixed into the second-stage data, which is common for mitigating catastrophic forgetting. We consider a replay ratio of 0.25, i.e., 25% of the second-stage data is from the first stage.
- **CPT from stable phase (sta):** We consider CPT starting from a base model checkpoint in the stable phase of the WSD learning rate schedule. This is to avoid adverse effects from re-warming the learning rate from a decayed value.

**Model growth.** In addition to depth-wise stacking described in the previous Section, we consider width expansion that double the size of the base model. For a growth factor of 2, we increase the size of the hidden dimension by $\sqrt{2}$ for width expansion. We further consider stacking variants:

- **Larger growth factor (x4):** We consider a larger growth factor of 4 times (instead of 2).
- **Model growth from stable phase (sta):** Similar to CPT, we train the stacked model starting from a base model checkpoint in the stable phase of the WSD learning rate schedule. This also avoids re-warming the learning rate from a decayed value.

**Comparing various functional forms.** Running the experiments as described above and obtaining the results, we fit the losses with the functional forms of Equations 6, 7, and 8. Following Hoffmann et al. (2022); Besiroglu et al. (2024), we perform optimization using the Huber loss ($\delta = 10^{-3}$) and

---

[3]As in Footnote 1, we omit the $+1$ term in $D + 1$ expressions for notational simplicity, such that terms proportional to $D_2$ are finite as $D_2 \to 0$.

the BFGS algorithm, to fit the logarithm of the loss via the LogSumExp trick applied to the RHS of functional forms. The leave-one-out root mean square error (RMS) serves as the goodness-of-fit metric.

As can be seen in Table 1, the multiplicative form with interaction, consistently achieves the lowest error across all methods and datasets, indicating that it best captures the underlying scaling behavior. To visualize these relationships, we compare the loss-versus-token plots of our proposed scaling law against alternative functional forms (Figure 3, left panel), using a 0.5B-to-1B model by stacking. The predicted losses from the alternative forms are visually confirmed to fit the empirical data worse. More importantly, the predicted losses from alternative forms with varying $D_1$ remain nearly parallel across the range of $D_2$, whereas ours converges to a crossover point, a distinct behavior that is not captured by other forms.

**Another variant of CPT scaling law.** When the dataset used for CPT is the same as the base model, we expect the scaling formula shown below holds:

$$L(D_1, D_2) = A(D_1 + D_2)^{-\alpha} + E \tag{9}$$

Hence, one may expect this holds for CPT on a different dataset as well, as assumed in Que et al. (2024); Wang et al. (2025). Fitting the above formula for the code (math) CPT scenario, we find that the RMS error is 0.0213 (0.0235), higher than the best ones in Table 1. Hence, our scaling law models CPT better, in contrast to previous assumptions, which may have overlooked the overtraining effects on the model.

One may wonder if the model growth methods also follow the above scaling law, since both stages use the same dataset. However, we find that the RMS error for (growth factor 2) expansion and stacking methods are 0.032 and 0.025 respectively, again larger than the ones in Table 1. Even though the same dataset is used in training, the model capacity is changed in model growth, which may explain why the above scaling law does not hold.

We further show that the loss follows a power law with respect to $D_1$ for fixed $D_2$ in Figure 3 (with more plots in Appendix D). This validates Condition 2 we impose for deriving the scaling laws.

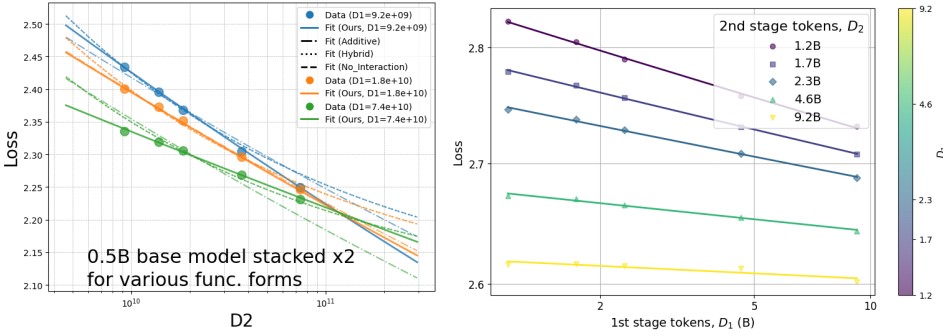

Figure 3: **Left: Visual comparison of various functional forms.** We show scaling behavior of second-stage training tokens ($D_2$) for different values of first-stage tokens ($D_1$) for model growth by stacking (growth factor 2, 0.5B-to-1B model), comparing various functional forms. Our proposed multiplicative scaling law with interaction (solid lines) fits the data (dots) best, and lines with different $D_1$ cross over, indicating saturation effects. Note that the hybrid functional form (dotted lines) nearly overlaps with the multiplicative without interaction form (dashed lines) in this plot. **Right: $D_1$ has power-law scaling.** We show scaling behavior of first-stage training tokens ($D_1$) for different values of second-stage tokens ($D_2$) for model growth by stacking (growth factor of 2), indicating that $D_1$ also has power-law scaling. More plots for other methods are available in Appendix D.

Overall, the proposed scaling law is robust across datasets, configurations, and methods investigated. While we acknowledge the possibility of edge cases that deviate from this law (e.g., methods that are fundamentally unscalable or perform poorly), our study is specifically focused on the practices considered scalable and representative of best practices. Consequently, we conclude that the scaling behavior and saturation effect identified here are representative of general bootstrapped pretraining methods employed in efficient LLM development.

Table 1: **Multiplicative scaling law with interaction consistently achieves lowest error.** Leave-one-out RMS error ($\times 10^{-3}$) for fitting the loss for bootstrapped pretraining of a 0.1B base model. Functional forms are from Equations 6 (and the specific case where $\alpha_3 = 0$), 7, and 8.

| RMS ($\times 10^{-3}$) | Code | Math | CPT (rep) | CPT (sta) | Exp x2 | Stk x2 | Stk x4 | Stk (sta) |
|---|---|---|---|---|---|---|---|---|
| **Mul.** | **1.573** | **1.737** | **0.987** | **1.371** | **2.790** | **2.845** | **2.152** | **2.957** |
| **Mul.** ($\alpha_3 = 0$) | 2.095 | 3.147 | 2.222 | 2.366 | 4.837 | 7.818 | 6.557 | 8.757 |
| **Add.** | 3.913 | 7.443 | 4.646 | 4.315 | 9.855 | 10.323 | 8.866 | 10.559 |
| **Hyb.** | 2.245 | 3.340 | 2.223 | 2.367 | 4.841 | 7.748 | 6.667 | 8.822 |

# 4 A CLOSER LOOK AT THE SCALING BEHAVIOR

## 4.1 EXPLAINING THE SATURATION PHENOMENON

Previous work like Lyle et al. (2023) has shown that the loss of neural networks' plasticity with training correlates with the increased sharpness of the loss landscape. Inspired by previous work which focuses on reinforcement learning tasks, we here study whether similar phenomena also occur in bootstrapped pretraining of LLMs.

To this end, we empirically investigate this phenomenon using gradient norms, which serve as a proxy for the geometry of the loss landscape (Zhao et al., 2022; Zhang et al., 2023; Xie et al., 2023). We show in Figure 4 the gradient norm curves of overtrained and well-trained models during bootstrapped pretraining. We find that overtrained models exhibit larger gradient norms, thereby indicating a sharper loss landscape that correlates with the observed saturation effect.

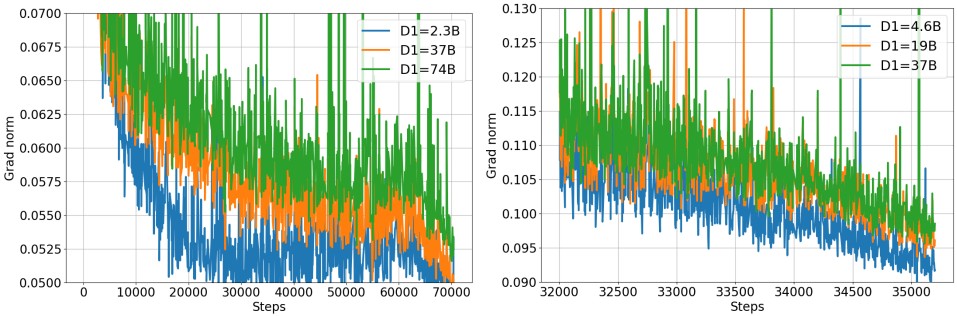

Figure 4: **Overtrained models have larger gradient norms when undergoing bootstrapped pretraining.** We show the gradient norm curves with respect to number of training steps in the second stage for base models trained with different values of first-stage tokens ($D_1$). We see that overtrained models have larger gradient norms, indicating that the loss landscape is sharper. **Left:** Continual pretraining on code data (1B model). **Right:** model growth by stacking (0.5B-to-1B model).

## 4.2 DIRECT STRATEGY TO MITIGATE SATURATION EFFECTS

Lyle et al. (2023) also showed that regularization, particularly layer normalization, is most effective at mitigating plasticity loss. However, layer normalization and other techniques (like weight decay and gradient clipping) are already employed in our models. Instead of pursuing further optimization or architecture modifications, we adopt a **direct, data-driven approach**: leveraging the derived scaling laws for both bootstrapped and from-scratch pretraining to guide practitioners in minimizing the negative effects of overtraining. We specifically target the stacking-based model growth scenario, as it exhibits more serious saturation effects and training from scratch is a common practical alternative.

Denote $L_{2N}^{\text{scratch}}(D)$ by the loss of a model of size $2N$ trained from scratch for $D$ tokens, and $L_N^{\text{growth}}(D_1, D_2)$ by the loss of a model of size $2N$ grown from a base model of size $N$ trained for $D_1$ tokens, and then trained for $D_2$ tokens.

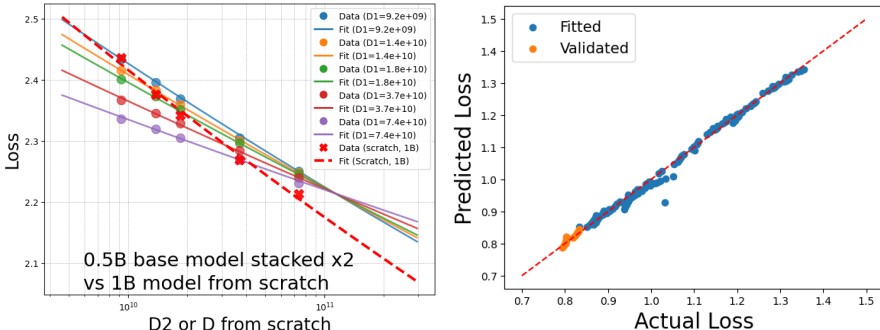

Figure 5: **Left: Loss versus second-stage (from-scratch) training token for stacking (training from scratch) a 0.5B-to-1B model (1B model)**. Data points and fitted lines using our multiplicative scaling law with different number of first-stage training tokens (power-law) are shown. It can be seen that as the number of training tokens increases, the losses from model growth saturate, and from-scratch training eventually outperforms it. **Right: Joint scaling law fit for continual pretraining on code data.** Orange points indicate the 10% of data with lowest losses used for validation.

In Figure 5 (left panel), we show the plots of the two losses, $L_N^{\text{growth}}(D_1, D_2)$, $L_{2N}^{\text{scratch}}(D)$ with $D = D_2$, using our experimental data as well as the fitted scaling laws for 0.5B-to-1B grown model and from-scratch 1B model training. We see that for small $D$ values, model growth outperforms from-scratch training for all $D_1$ values considered. However, as $D$ increases, the losses from model growth saturate, and from-scratch training eventually outperforms it. Consequently, the optimal choice between model growth and from-scratch training depends directly on the initial investment ($D_1$) and the total target training tokens ($D_2$ or $D$). This analysis provides a simple, scaling-law-based strategy for practitioners to mitigate the negative consequences of the saturation effect in bootstrapped pretraining.

### 4.3 JOINT SCALING INCORPORATING MODEL SIZE

We proceed to extend the data scaling law to include model size $N$. Let us first focus on CPT where the model size remains unchanged after bootstrapping. To determine the functional form, we impose that the loss follows the well-grounded "Chinchilla" scaling law (Hoffmann et al., 2022) (which jointly models the base model's loss with respect to dataset and model sizes) with respect to $D_2$ and $N$:

$$L_{D_1}(D_2, N) = A_{D_1} D_2^{-\alpha_{D_1}} + N_{D_1}^{-\beta_{D_1}} + E_{D_1}.$$

The straightforward functional form that satisfies the Chinchilla-style scaling law is:

$$L(D_1, D_2, N) = A D_1^{-\alpha_1} D_2^{-\alpha_2 + \alpha_3 \log D_1} + B N^{-\beta} + E, \qquad (10)$$

which is also consistent with the conditions (power-law ansatz) imposed in Section 3.

**Fitting the joint scaling law.** We conduct experiments with base models of sizes 15 million (M), 44M, 0.1B, 0.2B, 0.5B and 1B, training them with different numbers of first and second-stage tokens as in Section 3.2. To fit the joint scaling law, we use the same fitting procedure as in Section 3.2, additionally taking $N$ as an additional variable. In the left panel of Figure 5, we show that the formula fits the data well for CPT on code. See Table 7 in Appendix D.2 for the fitted coefficients. In the same Appendix, we further extend our results to model growth where model sizes increase.

Furthermore, the high goodness-of-fit to the validation data points confirms the law's predictive capability, allowing us to accurately extrapolate performance to models and datasets significantly larger than those studied experimentally (Figure 5, right panel).

By extrapolating the joint scaling law, we show in Appendix E.2 with a specific example that it can be used to predict the performance of larger models and datasets beyond those used in training. We can further study the compute optimality of CPT and model growth using the joint scaling law, as shown in Appendix E.3.

### 4.4 LIMITATIONS OF SCALING LAWS

A theoretical limitation of Equation 1 is that as $D_1$ increases, the effective scaling exponent for $D_2$, $\alpha_{\text{eff}} = \alpha_2 - \alpha_3 \log D_1$, decreases without bound, potentially becoming negative when $D_1 > e^{\alpha_2/\alpha_3}$. However, based on our fitted coefficients where $\alpha_2$ often exceeds $\alpha_3$ by at least an order of magnitude (see Table 7 in Appendix D.2), this theoretical threshold (e.g., $> 10^{17}$) is unlikely to be reached in practice. We anticipate the scaling law would break down due to physical limitations well before this point.

To formally model this inevitable thresholding behavior, which marks a deviation from the core power-law ansatz, we propose a modification motivated by Clark et al. (2022) (used for MoE scaling laws): In Equation 1, we replace $D_1$ with a saturated form, $\hat{D}_1$, defined as

$$\hat{D}_1 = \left(D_1^{-1} + D_{\max}^{-1}\right)^{-1}, \tag{11}$$

where $D_{\max}$ ($\lesssim e^{\alpha_2/\alpha_3}$) is a constant representing the maximum effective number of tokens that can be learned from the first stage. This modification is designed such that when $D_1 \gg D_{\max}$, $\hat{D}_1$ saturates to $D_{\max}$, preventing unbounded growth of the effective exponent.

We also notice a slight underestimation of the loss by the scaling law at very large (in terms of token-to-parameter ratio) $D_2$ values; see Figure 11 in Appendix E.1. This may be due to the limitation of the power-law ansatz, as also observed in Hoffmann et al. (2022). Otherwise, our scaling laws fit the data well across a wide range of $D_1$, $D_2$ and $N$ values studied.

## 5 CONCLUSION

We have provided a fairly broad scaling study of two-stage pretraining and there are many potential future directions worth diving deeper into. Let us highlight a few of them.

**Incorporating other factors into the scaling laws.** We have not considered the scaling behavior with respect to other method-specific factors, such as the CPT replay ratio (Que et al., 2024; Wang et al., 2025) and the growth factor in model growth (Du et al., 2024), as (1) our focus is on the more broadly applicable scaling behavior (i.e., scaling with respect to training tokens), and (2) adding more factors would increase the complexity of the scaling laws and the number of parameters to fit (and hence the amount of additional experiments/GPU hours/costs needed). We nevertheless expect that these factors can be incorporated quite naturally into our scaling laws which are power-law based.

**Eliminating saturation effects.** Although this work provides a mechanistic explanation and practical guidelines for navigating the saturation effects observed in bootstrapped pretraining, the fundamental question of how to better mitigate these effects remains open. Highly scalable regularization techniques commonly employed in LLM training (e.g., layer normalization) do not appear to fully eliminate the issue. Developing more advanced and scalable techniques to achieve this would provide a crucial solution, complementary to the scaling-law-based guidelines presented here.

**Theory.** Providing a theoretical origin of the scaling laws is an important open question in the field, but existing work has been focusing on relatively simple theoretical setups like *power-law random feature models* to reproduce power-law scaling in deep neural networks (see Paquette et al. (2024) and references therein). To our best knowledge, there is no existing theory that can explain the change of scaling exponents with respect to the amount of pretraining data. In the current work, we have focused on the practical aspects of the scaling laws, following previous influential scaling law work (Kaplan et al., 2020; Hoffmann et al., 2022), and have not attempted to provide a theoretical explanation of the scaling laws as the current theoretical understanding of scaling laws is still limited. It would be interesting to study if such a change of (effective) scaling exponents can be explained by modifying existing theories minimally or if new theoretical frameworks are needed.

## REPRODUCIBILITY STATEMENT

We include data and notebooks for reproducing the main results and plots in the supplementary material. Example scripts for training the models are also provided. We further include detailed experimental configurations and model architectures in Appendix B.

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

## A  Notations

We summarize main notations used in the paper.

- $L$: Cross-entropy loss or the (natural) logarithmic loss
- $D$: Dataset size in token
- $N$: Non-embedding model size or number of non-embedding parameters
- $A, B, F$: Scaling factors of the power law, independent of the variable under consideration
- $E$: Irreducible loss of the power law, independent of the variable under consideration
- $\alpha, \beta, \gamma$: Scaling exponents of the power law, independent of the variable under consideration
- $n_{\text{layer}}$: number of layers of a model
- $d_{\text{model}}$: hidden dimension size of a model
- $d_{\text{MLP}}$: intermediate hidden dimension size of a model

## B  More on Architecture and Experimental Design

### B.1  Megatron-LM Configuration

**Infrastructure.** Our experiments are performed on multiple nodes, each consisting of 8 NVIDIA H100 80 GB GPUs, interconnected via InfiniBand HDR. The software we use for training is the Megatron-LM library Shoeybi et al. (2019).

We use and modify the Megatron-LM (core v0.8.0) library for our experiments[4]. Models are trained with data type bfloat16. Other optimization libraries used include FlashAttention Dao et al. (2022) and TransformerEngine[5]. See the example scripts provided in supplementary material.

### B.2  Model Configuration

Let us elaborate more on our model configuration. The intermediate hidden dimension size, $d_{\text{MLP}}$, is set to be four times the hidden dimension size, i.e., $4d_{\text{model}}$. Bias is not used in the linear layers. We do not consider efficiency-motivated implementations like grouped query attention as well. Attention head number is chosen to increase with model size following standard practices. Other designs of the architecture follow Llama2's closely (Touvron et al., 2023).

We vary model sizes keeping the ratio $n_{\text{layer}}/d_{\text{model}}$ to lie in the range 32 to 64, as in Kaplan et al. (2020). Smaller models for used for ablation studies. See Table 2 for the exact numbers for the model configuration.

Table 2: Models used in our study and their parametric details. Note that $d_{\text{MLP}}$, is set to be $4d_{\text{model}}$.

| Model | $n_{\text{layer}}$ | $d_{\text{model}}$ | $n_{\text{head}}$ | $N$ |
|-------|------|------|------|------|
| **15M** | 9 | 320 | 4 | 14,751,680 |
| **44M** | 12 | 480 | 8 | 44,248,800 |
| **0.1B** | 15 | 640 | 8 | 98,323,840 |
| **0.2B** | 21 | 832 | 8 | 232,623,040 |
| **0.5B** | 26 | 1,120 | 16 | 521,889,760 |
| **1B** | 30 | 1,504 | 16 | 1,085,859,424 |

### B.3  Training Configuration

As discussed in the main text, we adopt the WSD learning rate schedule for all experiments. The number of warmup steps is set approximately equal to the model size, as suggested by Porian et al. (2024). In the final phase of training, the learning rate decays linearly to 10% of its peak value, with the decay phase spanning roughly 10% of the total training steps, following the setup in Hägele

---

[4]https://github.com/NVIDIA/Megatron-LM
[5]https://github.com/NVIDIA/TransformerEngine

Table 3: Training configuration used throughout the paper.

| Configuration | Details |
|---|---|
| Context length | 1,024 |
| Embedding tying | False |
| Optimizer | AdamW Loshchilov et al. (2017) |
| Adam $\beta_1$ | 0.9 |
| Adam $\beta_2$ | 0.95 |
| Adam $\epsilon$ | 1e-8 |
| Weight decay | 0.1 |
| Gradient clipping | 1.0 |

et al. (2024). To emulate varying token budgets, we save intermediate checkpoints at logarithmically spaced intervals.

The general training configuration is summarized in Table 3, while model-specific hyperparameters, such as warmup iterations, initialization standard deviation ($\sqrt{2/5d_{\mathrm{model}}}$ (Le Scao et al., 2022)), maximum training steps, batch size, and tuned learning rate, are provided in Table 4. Batch size is scaled with model size according to standard practice, without tuning for optimality. The number of tokens is increased with model size as well, such that the parameter-to-token ratio remains roughly constant across different model sizes. We note that no specialized techniques for mitigating training instabilities, such as Z-loss or QK normalization, are employed, as such instabilities do not arise in our experiments.

Table 4: **Model-dependent training configuration**. "init. size" refers to the standard deviation of the normal distribution used for initializing the weights. "Std iter." refers to the number of iteration run on the model to reach about 70 times the model size in tokens, the standard training length used in the scaling law experiments. For several models, we also run longer training for exploration.

| Model | warmup iter. | init. size | Std iter. | batch size | LR |
|---|---|---|---|---|---|
| **15M** | 200 | 0.035 | 17,600 | 128 | 8e-3 |
| **44M** | 200 | 0.029 | 17,600 | 256 | 4e-3 |
| **0.1B** | 200 | 0.025 | 17,600 | 512 | 4e-3 |
| **0.2B** | 400 | 0.022 | 35,200 | 512 | 2e-3 |
| **0.5B** | 800 | 0.019 | 70,400 | 512 | 4e-4 |
| **1B** | 800 | 0.016 | 70,400 | 1024 | 4e-4 |

## B.4 ON THE CHOICE OF DATA DOMAINS STUDIED

Here, we argue that our experimental setup for CPT incorporates a significant domain shift, and it is highly relevant to frontier model research.

- **Domain gap in our setup.** The base models are purposefully pretrained exclusively on the Common Crawl portion of the Slimpajama-DC dataset. This explicitly excludes data heavily skewed toward specialized domains such as mathematics (like arXiv) or code (like dedicated GitHub repositories), thereby maximizing the domain gap for the subsequent CPT stage.

- **Small overlap of domains.** While some minimal token overlap between the generic web corpus and the specialized domains is inevitable, it is quantitatively small. We estimate that math/code data accounts for approximately 0.3 percent of the total Slimpajama-DC corpus. [6] Consequently, the domain shift in our CPT experiments, which move to dedicated code and math corpora, is substantial.

- **Relevance to frontier LLM training.** The strategy of pretraining mainly on generic web data and subsequently applying CPT on specialized domains like code (Stack/StarCoder) and mathematics (OpenWebMath) is a standard and highly relevant practice for state-of-the-art LLMs.

---

[6]Estimated based on filtering performed for the OpenCoder (Huang et al., 2025) dataset, which extracted math/code data from a similar CommonCrawl-derived corpus, FineWeb (Penedo et al., 2024).

- **Model growth.** For model growth experiments, the primary goal is to isolate and quantify the effect of increasing model capacity, not domain shift. Therefore, the second stage is intentionally conducted on the same generic web dataset as the base model, allowing us to accurately quantify the scaling behavior of the growth technique itself.

To summarize, our current findings are robust within the most practically relevant and challenging domain-transfer scenarios currently employed for large-scale LLM training.

### B.5 GPU Hours and Costs

Instead of reporting the actual runtimes on our cluster, which varied in our experiments due to many factors affecting the cluster (number of available nodes, congestion, etc.), we give a theoretical estimate of total GPU hours used for obtaining the joint scaling law, which involves running the largest tested model with most training tokens in this paper.

The estimate is as follows. We calculate the FLOPs for training the largest first and second-stage models with maximum iterations using the $6ND$ approximation, ignoring the additional FLOPs required to continued pretrain models with shorter iterations (as we can reuse the intermediate checkpoints). [7] We further assume that the per-second TFLOPs of the GPU is 400. We obtain around $10^9$ TFLOPs. Taking into account of additional experimentation and ablation runs, we estimate that around 3000 GPU hours were used for the entire project. Assuming a cost of 2 USD per GPU hour, the total cost is around 6000 USD.

**Scaling law methodology and cost justification.** We note that our study contains over 450 total runs, as accurately mapping the relationship between multiple scaling variables (model size $N$, first-stage tokens $D_1$, and second-stage tokens $D_2$) necessitates a multitude of controlled experiments on models of various sizes. This methodology contrasts with single, large-scale training runs (e.g., training one 7B model from scratch), which would provide fewer data points for fitting complex scaling laws. This rigorous approach, balanced with the need for a reasonable overall budget, dictates our focus on models up to 1B parameters for extensive experimentation.

## C Training Details

### C.1 Ablation of Learning Rate Schedules

Here, we compare the performances of using WSD and the commonly used learning rate (LR) cosine schedules (decaying to 10% of the peak LR value) (Touvron et al., 2023). The model size we use is 0.1B, with the following training configuration: batch size 512, 4,000 training iterations, and 200 warmup iterations. We can see from Figure 6 that both schedules yield similar (with WSD achieving slightly better final loss) performances. This justifies our choice of using the WSD schedule throughout the paper.

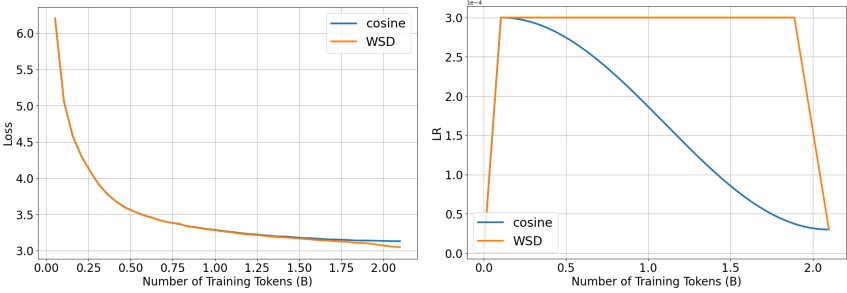

Figure 6: **WSD vs cosine LR schedule.** We show that the WSD LR schedule achieves similar (even slightly better) final loss compared to the more commonly used cosine LR schedule.

---

[7]We have more than 450 runs; estimated from 25 data points for each of the 8 methods considered in Table 1, and 125 data points for the two joint scaling law experiments (CPT and model growth).

## C.2 ABLATION OF DATA REPETITION

During model growth, we have used the same data as the base model for the second stage of training. This is because the base model is often trained with a large amount of data, and it is not always possible to obtain a large amount of additional data. However, this means that the second stage of training involves repeating the same data. To study the effect of data repetition, we have made an ablation study, where we compare the performance of using the same data as the base model and using different portion of the Slimpajama data. As shown in Figure 7, we find that the difference is small.

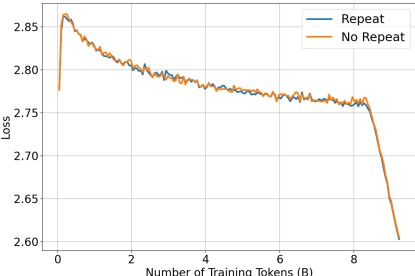

Figure 7: **Ablation of data repetition during model growth.** We compare the performance of using the same data as the base model and using different data for the second stage of training. The difference is small, indicating that data repetition does not affect the scaling behavior significantly.

## C.3 EVALUATION WITH STANDARD BENCHMARKS

We compare the performance of our trained 1B base model against existing models with similar sizes, Pythia (Biderman et al., 2023) and TinyLlama (Zhang et al., 2024b), based on standard natural language processing benchmarks, ARC (Clark et al., 2018), lambada (Paperno et al., 2016), logiqa (Liu et al., 2021), piqa (Bisk et al., 2020), sciq (Welbl et al., 2017), and winogrande (Sakaguchi et al., 2021).

Table 5 shows the results. We see that the model perform similarly to the open models, indicating that our models have been trained correctly.

Table 5: **Benchmarks' performance comparison across models.** Reported scores are accuracies (normalized by byte length whenever applicable). The first two columns are scores of existing models. The last column is evaluation results of the largest base model trained with the most number of tokens in this work. Our models are evaluated with the LM Evaluation Harness v0.4.0 library (Gao et al., 2024).

| Models | Pythia-1B | TinyLlama-1.1B | Our 1B model |
|---|---|---|---|
| **Datasets** | Pile | Slimpajama & Starcoder | Slimpajama |
| **Tokens** | 100B | 103B | 74B |
| ARC-c | 25.59 | 24.32 | 27.65 |
| ARC-e | 47.26 | 44.91 | 52.10 |
| lambada | 53.52 | - | 45.08 |
| logiqa | 29.49 | - | 26.11 |
| piqa | 69.31 | 67.30 | 65.89 |
| sciq | 77.3 | - | 78.10 |
| winogrande | 51.22 | 53.28 | 54.93 |
| **Avg.** | 50.53 | - | 49.98 |

### C.4 CONTINUAL PRETRAINING DETAILS

Using a lower LR for CPT is also a common practice (Ibrahim et al., 2024). As an ablation, we have experimented with using a constant LR schedule setting the value to be the final LR for the first-stage base model training, on a 0.1B model. We find that it yields worse performance (see Figure 8). Hence, we use the same LR (and LR schedule) as the base model for CPT.

### C.5 MODEL GROWTH DETAILS

For completeness, we provide more details on the model growth methods used in this work.

**Width expansion details.** Let us elaborate more on the width expansion method used in this work. We first give a more precise defintion of function preservation. Let $F$ be a function and $G$ as the growth operator. The function preservation condition is defined as:

$$F(x) = G(F)(x), \forall x \in \mathcal{X}$$

where $\mathcal{X}$ is the input space.

When performing width expansion, the neuron values of each layer are expanded by duplicating the weights of existing neurons, and dividing the output weights by the growth factor.

**Stacking details.** We first note that stacking does not preserve function but is empirically found to work well in practice (Du et al., 2024). We use the recommended stacking procedure in Du et al. (2024), which is, letting $M$ be the non-embedding part of the base model, the stacked model with growth factor $k$ is given by:

$$M' = M \circ M \circ ... \circ M \quad (k \text{ times})$$

where $\circ$ denotes function composition. The embedding and final layer are then simply copied from the base model.

**Low LR.** We also experiment with using a lower constant LR for stacking as above, and find that it also yields worse performance (see Figure 8). Hence, we use the same LR (and LR schedule) as the base model for stacking as well.

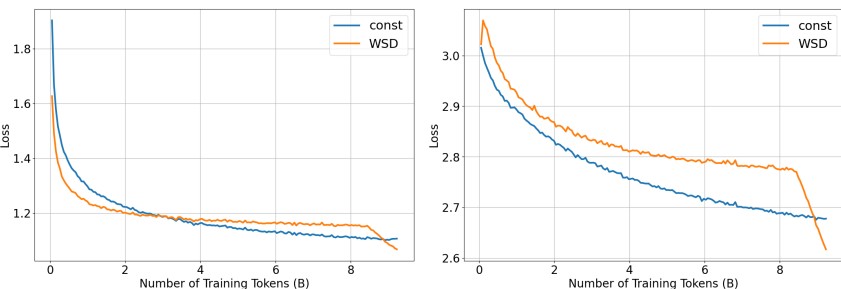

Figure 8: **Bootstrapped pretraining with lower LR.** We show that using the same LR as the base model achieves better final performance than using a lower LR for a 0.1B model continual pretrained on code data (left) and stacking (right).

## D SCALING LAWS DETAILS

### D.1 FITTED EXPONENTS AND OTHER RESULTS

We show the fitted exponents of the multiplicative scaling laws studied in Table 1 in Table 6.

We further show that $D_1$ has power-law scaling in Figure 9, for CPT on mathematics data and model growth by expansion, which justifies the multiplicative form of the scaling laws.

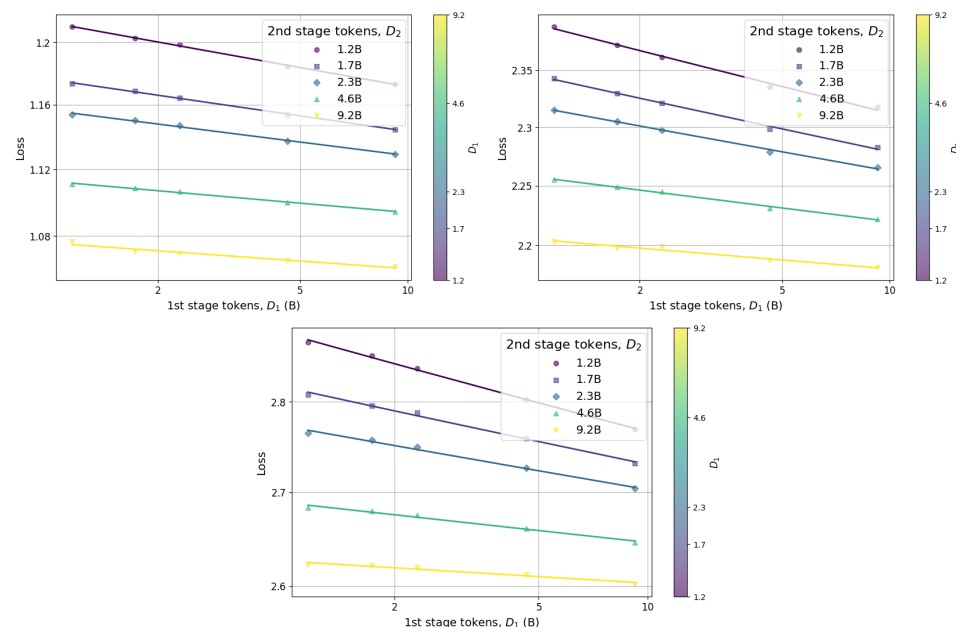

Figure 9: $D_1$ **has power-law scaling.** We show scaling behavior of fist-stage training tokens ($D_1$) for different values of second-stage tokens ($D_2$), indicating that $D_1$ also has power-law scaling. **Top left:** Continual pretraining on code. **Top right:** Continual pretraining on mathematics data. **Bottom:** model growth by expansion (growth factor 2).

Table 6: **Fitted exponents for the multiplicative scaling law.** Corresponding $\alpha_1, \alpha_2, \alpha_3$ values for the multiplicative interaction fits shown in Table 6.

| Variant | $\alpha_1$ | $\alpha_2$ | $\alpha_3$ |
|---|---|---|---|
| **CPT (code)** | 0.106 | 0.146 | 0.004 |
| **CPT (math)** | 0.981 | 0.388 | 0.017 |
| **Expand** | 0.549 | 0.852 | 0.024 |
| **Stack** | 0.515 | 0.350 | 0.017 |
| **CPT (replay)** | 0.424 | 0.626 | 0.018 |
| **CPT (stable)** | 0.920 | 0.156 | 0.004 |
| **Stack (x4)** | 0.644 | 0.829 | 0.028 |
| **Stack (stable)** | 0.891 | 0.507 | 0.009 |

## D.2 JOINT SCALING LAW

For model growth, we also fit the joint scaling law in Equation 10. Note that there is ambiguity in defining the model size $N$ in the context of model growth. We consider $N$ to be the size of the model before growth, and as we keep the growth factor fixed to be 2 in our experiments, the new model size after growth is $N' = 2N$. Therefore, $N$ and $N'$ differ by a constant factor of 2, which can be absorbed into the coefficient $B$ in Equation 10. Unless stated otherwise, we use $N$ in the fitting of the joint scaling law for model growth. In Figure 10, we show the fit of the joint scaling law for stacking.

We provide the fitted coefficients of the joint scaling law in Table 7. In addition to bootstrapped pretraining, we also show the fitted coefficients for base models trained from scratch on the same dataset as the second stage (code data for CPT, and the same data as the base model for model growth). For model growth, we consider a growth factor of 2, and fit the parameters with model size $N$ before stacking for comparison conveniences. We further note that the fitted coefficients are produced by fitting the joint scaling law to *all* data points collected, including those used for validation in Figure 10.

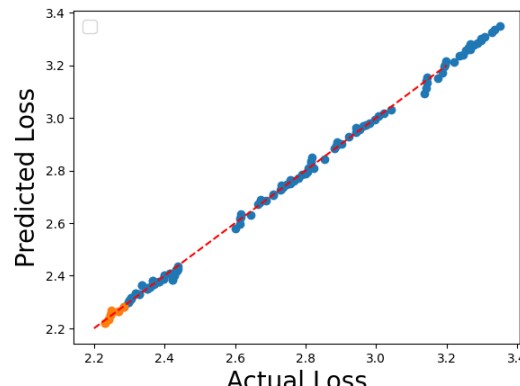

Figure 10: **Fits of the joint upcycling scaling law of stacking.** 10% of the collected data points with lowest losses are used for validation (orange points).

Table 7: **Fitted coefficients for joint scaling laws.** Note that for stacked models, we fit the coefficients with model size $N$ before stacking for comparison conveniences.

|  | $\alpha/\alpha_1$ | $\alpha_2$ | $\alpha_3$ | $\beta$ | $A$ | $B$ | $E$ |
|---|---|---|---|---|---|---|---|
| **Base (Slimpajama)** | 0.092 | - | - | 0.105 | 10.383 | 10.085 | 0.041 |
| **From-scratch (code)** | 0.113 | - | - | 0.234 | 8.143 | 27.286 | 0.105 |
| **CPT (code)** | 0.048 | 0.126 | 0.001 | 0.238 | 15.062 | 27.234 | 0.105 |
| **Stack** | 0.087 | 0.119 | 0.003 | 0.173 | 33.394 | 22.471 | 0.041 |

### D.3 WHY FITTING SCALING LAWS SEPARATELY?

We justify our decision to fit two separate scaling laws—one for models trained from scratch and another for bootstrapped pretraining—instead of employing a unified formulation that spans all stages.

First, the scaling behavior of models trained from scratch is expected to differ from that of models undergoing bootstrapping. Specifically, at the limit $D_1 \rightarrow 0$, grown models are initialized from a pretrained base model, whereas scratch-trained models start from random weights. This difference in initialization leads to distinct learning dynamics and, consequently, different scaling law parameters.

Similarly, the scaling behavior of the base model differs from that of the second-stage training in the limit $D_2 \rightarrow 0$. As shown in Figure 8, second-stage training often begins with a *rewarming* phase, during which the loss initially increases before decreasing. This early instability deviates from the expected scaling of dense models, although the overall trend remains correlated—supporting the validity of Condition 1, due to function preservation and empirical observations that rewarming does not entirely disrupt loss behavior (as quoted from Clauset et al. (2009): "In practice, few empirical phenomena obey power laws for all values of $x$").

Finally, our preliminary experiments indicate that a unified scaling law does not provide a satisfactory fit across both stages. We therefore opt to model them separately. We also note that there may exist alternative functional forms that could better capture the full range of behavior, but they may violate some of the well-established conditions for scaling laws (power law), and may overcomplicate the analysis or overfit the data; hence, we leave their exploration to future work.

## E MORE OBSERVATIONS AND IMPLICATIONS OF THE SCALING LAWS

### E.1 TRAINING WITH EXTREMELY LARGE SECOND-STAGE TOKEN-TO-PARAMETER RATIO

Our scaling laws predict that base models trained with different number of tokens in the first stage ($D_1$) will converge at some point, and diverge again afterwards, with smaller $D_1$ models performing better at large $D_2$. To see if the latter occurs, we train a 15M model with different $D_1$ values, stack them with growth factor 2, and continue training with extremely large number of tokens in terms of token-to-parameter ratio (up to around a factor of 1300 or 20B tokens).

Figure 11 shows the results for this stacking scenario. Consistent with the predictions of our scaling law, the fitted lines demonstrate that the losses converge, and beyond the crossover point, models with smaller $D_1$ achieve lower losses.

We note, however, that at the largest tested $D_2$ values, a slight underestimation of the loss by the fitted scaling law is observed for some data points. This subtle deviation occurs at extremely high token-to-parameter ratios and suggests a limitation of the simple power-law ansatz in fully capturing the scaling behavior in this extreme regime. This coincides with an observation made by Hoffmann et al. (2022), where there is a "curvature" in loss-versus-compute plots at very high compute values. This also means that the saturation effects are potentially even stronger in this extreme regime than predicted by our scaling law. Nevertheless, the overall trend confirming the saturation and eventual divergence remains clear.

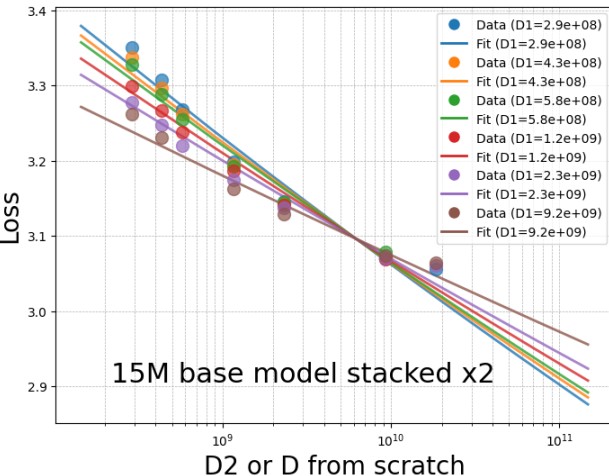

Figure 11: **Stacking with extremely large token-to-parameter ratio.** We continue training 15M-to-30M stacked models with different first-stage training tokens ($D_1$) to extremely large second-stage token-to-parameter ratios (up to around 1300). The data points largely follow the predicted scaling laws (solid lines). At the largest $D_2$ experimented with, the losses highly overlap with each other, but still show a trend of $L(D_1 = 2.9\text{e}{+}08, D_2) > L(D_1 = 2.3\text{e}{+}09, D_2) > L(D_1 = 9.2\text{e}{+}09, D_2)$, which is predicted by our scaling law predicting that models with smaller $D_1$ achieve lower losses in this region. Underestimation of the loss by the fitted scaling law is observed at the largest $D_2$, indicating a limitation of the power-law ansatz in this extreme regime.

### E.2 IMPLICATION BASED ON SCALING LAWS' EXTRAPOLATION

We consider an implication based on the extrapolation of the scaling laws, motivated by studies on MoE upcycling (Komatsuzaki et al.; Liew et al., 2025).

A key practical consideration is how much of the *initial investment* in pretraining, the so-called *sunk cost* ($D_1$), can be effectively leveraged in the second stage of training. This question has been explored in the context of MoE upcycling strategies. For example, it was found that upcycling yields benefits up to 120% of the sunk cost. That is, to match the performance of an upcycled MoE model that underwent an additional 0.4 trillion tokens of training after an initial 2T tokens, training a comparable MoE from scratch would require 2.4T tokens—representing an effective saving of 2T tokens.

We investigate this in model growth by stacking (growth factor 2). We define $D^*$ as the number of tokens required for training from scratch to match the performance of a grown model with the same sunk cost, following Liew et al. (2025):

$$L_{2N}^{\text{scratch}}(D^*) = L_N^{\text{grown}}(D_1 = D^*, D_2 = D^*) \tag{12}$$

where $L_{2N}^{\text{scratch}}(D)$ is the loss of a model of size $2N$ trained from scratch for $D$ tokens, and $L_N^{\text{grown}}(D_1, D_2)$ is the loss of a model of size $2N$ grown from a base model of size $N$ trained for $D_1$ tokens, and then trained for $D_2$ tokens. To obtain $L_{2N}^{\text{scratch}}(D)$, we fit the losses of models trained from scratch with the Chinchilla-style scaling law (see Table 7 in Appendix D.2 for the fitted coefficients).

The right panel of Figure 12 shows that $D^*$ decreases with increasing model size, with $D^*$ equal to 13T tokens for a 100B model. When $D_2 \lesssim D^*$, the required from-scratch tokens to catch up is more than 100% of $D_1$: model growth remains more efficient than training from scratch. However, beyond this threshold, from-scratch training becomes more efficient. Solving Equation 12, we can approximate the threshold $D^*$ analytically as $D^* \simeq 13 \left(\frac{N}{10^{11}}\right)^{-0.6 - 0.04 \log(N/10^{11})}$ T tokens.

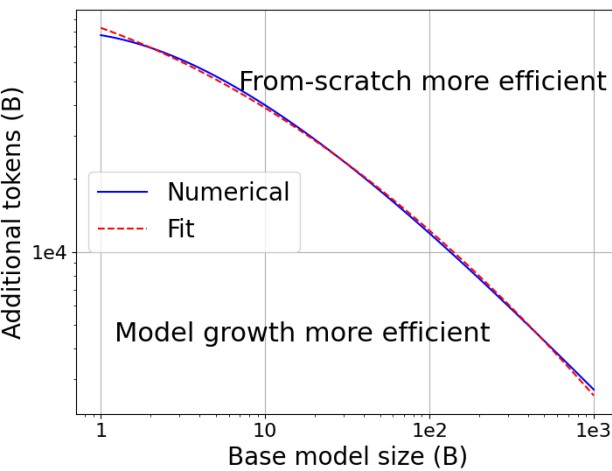

Figure 12: **Model growth efficiency decreases with sunk cost and model size.** For token budgets above the curve(s), training from scratch is more efficient than stacking-based model growth (growth factor 2); for budgets below, model growth remains advantageous. Shown are the numerical (blue) and analytical (red) solutions of Equation 12. Vertical axis is $D^*$ as in Equation 12.

### E.3  Compute Optimality

For scaling studies of LLMs, the cost of training is often estimated using floating point operations (FLOPs), which we simply refer to as compute. The compute cost can be approximated as $C = 6ND$ (Kaplan et al., 2020). One is often interested in training LLMs with the least amount of compute. For CPT, optimizing the compute based on Equation 6 leads to the scaling relations:

$$D_2^{\text{opt}} \propto C_2^{\frac{\beta}{\beta + \alpha_{\text{eff}}}}, \quad N^{\text{opt}} \propto C_2^{\frac{\alpha_{\text{eff}}}{\beta + \alpha_{\text{eff}}}} \tag{13}$$

where $\alpha_{\text{eff}} := \alpha_2 - \alpha_3 \log D_1$. Notably, as $D_1$ increases, $\alpha_{\text{eff}}$ decreases, meaning larger models require more tokens for compute-optimal bootstrapped pretraining.

For model growth with growth factor 2, we want to scale $N_2, D_2$ optimally, $N_2^{\text{opt}}, D_2^{\text{opt}}$, given a FLOPs budget and fixing $D_1$, while minimizing the loss $L$, which we write as $L_{D_1}(D_2, N_1)$. Here, $N_1$ is the model size before growth, and $N_2 = 2N_1$. This is equivalent to solving the following:

$$\frac{\partial}{\partial D_2} L_{D_1}(D_2, C_2/12D_2) \bigg|_{D_2 = D_2^{\text{opt}}} = 0,$$

$$\frac{\partial}{\partial N_1} L_{D_1}(C_2/12N_1, N_1) \bigg|_{N_1 = N_1^{\text{opt}}} = 0$$

where we have used $N_2 = 2N_1$ and $C_2 = 6N_2 D_2$. Solving the above equations leads to

$$D_2^{\text{opt}} = G \left( \frac{C_2}{12} \right)^a,$$

$$N_1^{\text{opt}} = G^{-1} \left( \frac{C_2}{12} \right)^b$$

where

$$G := \left( \frac{A_{\text{eff}} \alpha_{\text{eff}}}{B\beta} \right)^{1/(\alpha_{\text{eff}} + \beta)}$$

$$a := \frac{\beta}{\alpha_{\text{eff}} + \beta}$$

$$b := \frac{\alpha_{\text{eff}}}{\alpha_{\text{eff}} + \beta}$$

$$A_{\text{eff}} := A D_1^{-\alpha_1}$$

$$\alpha_{\text{eff}} := \alpha_2 - \alpha_3 \log D_1$$

We can henceforth relate $D_2^{\text{opt}}$ and $N_1^{\text{opt}}$ via

$$D_2^{\text{opt}} = G \left( G N_1^{\text{opt}} \right)^{a/b} \propto \left( N_1^{\text{opt}} \right)^{\beta/\alpha_2 - \alpha_3 \log D_1}$$

and

$$N_1^{\text{opt}} = G^{-1} \left( G^{-1} D_2^{\text{opt}} \right)^{b/a} \propto \left( D_2^{\text{opt}} \right)^{(\alpha_2 - \alpha_3 \log D_1)/\beta}$$

Notably, as $D_1$ increases, $\alpha_{\text{eff}}$ decreases, meaning larger models require more tokens for compute-optimal bootstrapped pretraining. We leave empirical verification of these relationships to future work, primarily due to cost considerations (see Appendix B.5 for an estimate of GPU hours spent in the work).

## F  THE USE OF LARGE LANGUAGE MODELS (LLMS) IN THIS WORK

We have used LLMs solely for proofreading and improving the readability of the paper.

