# OpenReview forum: "From Acceleration to Saturation: Scaling Behavior of Bootstrapped Language Model Pretraining"
_ICLR.cc/2026/Conference — Submitted to ICLR 2026_

### Official Review · Reviewer_5caV · 2025-10-26

**Soundness:** 2
**Presentation:** 3
**Contribution:** 3
**Rating:** 4
**Confidence:** 3

**Summary:**

The paper investigates the scaling behavior of bootstrapped language-model pretraining. The authors discover a predictable saturation effect: the scaling exponent with respect to second-stage tokens D_2 decays logarithmically with first-stage tokens D_1. They introduce a multiplicative scaling law which accurately captures this interaction across model sizes and bootstrapping variants. Extensive fits show their law outperforms additive or hybrid forms and remains valid under different learning-rate schedules, replay ratios, and growth factors. By pairing the law with Chinchilla-style compute scaling, they derive practical thresholds that indicate when training from scratch overtakes bootstrapping, offering actionable guidance for compute-optimal pretraining.

**Strengths:**

1. This paper is the first to identify and quantify the saturation effect in bootstrapped pretraining, providing a theoretically grounded scaling law with an interaction term that precisely models the phenomenon.

2. Large-scale empirical validation across diverse model sizes, training strategies, and configuration variants demonstrates the robustness and general applicability of the proposed scaling law.

3. The work extends the data-only scaling law to a joint data-and-model-size formulation, delivering practical criteria for choosing between bootstrapping and from-scratch training under compute budgets.

**Weaknesses:**

1. Limited Scale of Experiments: The study's largest model is 1.1B parameters, trained on hundreds of billions of tokens, which is significantly smaller than modern LLMs (often 7B to trillions of parameters). The observed scaling saturation may not hold at these larger, more relevant scales, limiting the practical applicability of the proposed laws for state-of-the-art model training.

2. Narrow Scope of Domain Shift: The analysis of saturation is primarily conducted on domains (e.g., code, math) that are relatively similar to the original pre-training data, especially in the model growth experiments. The work does not investigate if the same saturation effects and scaling laws apply to highly dissimilar domains or cross-modal tasks, raising questions about the generalizability of the findings.

3. Lack of Mechanistic Explanation: The paper provides a descriptive scaling law and a high-level hypothesis about loss of plasticity but lacks a deeper investigation into the underlying cause of the saturation. Without analyzing the mechanistic reasons, the work offers limited guidance on how to potentially mitigate the saturation effect.

**Questions:**

As described in Weakness. Expanding the experimental scope to include larger models (7B+) and more rigorous cross-domain benchmarks would greatly enhance the generalizability and broader impact of the findings.

---

> ### Author Response · Authors · 2025-11-21
> **Reply 1**
>
> Thank you for the thoughtful review of our work. We are especially grateful for the reviewer's **positive evaluation of our first identification and quantification of the saturation effects, large-scale empirical validation, and joint data-and-model-size formulation delivering practical criteria for pretraining** in our work.
> In the following, let us clarify comments raised by the reviewer.
>
> ## Comments on Weaknesses
> > Limited Scale of Experiments
>
> We acknowledge the concern about generalizability to larger models; however, we respectfully argue that our current experimental scale is sufficient to validate our core findings and establish the proposed scaling laws.
> - We showed that **the scaling behavior is reliable and predictable**: We validated the joint scaling law across a range of model sizes, from 15M to 1B in our study. Figure 4 (Left) further showed empirically that **sub-0.5B models can reliably predict the loss behavior at the larger 1B scale**. This provides strong empirical justification for the robustness of our scaling laws. The scaling relationships derived from this validated space are then used to predict the behavior of much larger models (7B or more). This is the standard, accepted methodology in scaling law research, which allows researchers to inform the training of models far larger than those used in the original scaling study (e.g. Chinchilla paper).
> - **Budget constraints**: We note that our 1B parameter model already consumed a substantial portion of our available computational resources (estimated in Appendix B.4) as it requires extensive grid search (we need to train many times on different models). Increasing by a single order of magnitude in both model size and dataset size would require multiple times the total GPU budget we have already spent, **estimated to be more than tens of thousands of USD. 7B+ models with trillions of tokens are production-scale models which cost close to sub-one million USD**. Under the constraints of an academic research setting, such an expenditure is not feasible.
> - Previous influential work like the OpenAI scaling law paper also studied models only up to the sub-1B scale. Therefore, we believe it is reasonable to run multiple experiments (we have over **450 runs** in this paper) on model sizes up to this scale.
>
> > Narrow Scope of Domain Shift
>
> Thank you for raising this point. We believe there may be a misunderstanding regarding the similarity of the domains used in our CPT experiments and our original base pretraining data. We argue that our experimental setup already employs a significant domain shift and that our findings directly impact current frontier model research.
> - The base model was pretrained exclusively on the CommonCrawl portion of the Slimpajama-DC dataset. This purposefully excludes data heavily skewed toward math (like arXiv) or code (like the dedicated GitHub portion), thereby maximizing the domain gap for the CPT stage.
> - While some token overlap between the generic web corpus and specialized domains is inevitable, it is quantitatively small. **We expect math/code data accounts for approximately 0.3 percent of the total** (estimated based on the OpenCoder dataset, which filters math/code data from FineWeb, a dataset extracted from CommonCrawl, similar to Slimpajama). Therefore, the domain shift in our CPT experiments is substantial and not "relatively similar" as suggested.
> - Moreover, the strategy of pretraining mainly on web data and subsequently applying CPT on specialized domains like math and code is a **standard and highly relevant practice for frontier models such as DeepSeekMath and DeepSeek-Coder**. Our observed saturation effects in these domains thus have a direct and immediate impact on the efficiency and design of these current and future LLMs.
> - For model growth experiments, the major goal is to increase the model capacity; **the second stage is intentionally conducted on the same generic web dataset as the base model**. This isolates the effect of capacity scaling from domain shift, allowing us to accurately quantify the scaling behavior of the growth technique itself.
> - Our current findings **are robust within the most practically relevant and challenging domain-transfer scenarios for LLMs**.

---

> > ### Author Response · Authors · 2025-11-21
> > **Reply 2**
> >
> > > Lack of Mechanistic Explanation
> >
> >  Thank you for this constructive feedback.
> > We have since conducted a deeper investigation into the underlying causes of the saturation effect and **updated our manuscript to include these findings** (see Appendix F in the updated attached manuscript).
> >
> > We hypothesize that the scaling saturation is related to the **sharpness of loss landscape in overtrained base models**.
> > We have empirically investigated this phenomenon using gradient norms, which serve as a proxy for the geometry of the loss landscape.
> > - We show in Figure 12 that overtrained base models (large $D_1$) **exhibit larger gradient norms during the second stage of bootstrapped pretraining compared to models that were less trained** (small $D_1$).
> > - A larger gradient norm is widely associated with a sharper loss landscape. This mechanistically explains our main observation of the paper: the sharper landscape restricts the model's ability to efficiently move towards a new, lower-loss minimum during the second stage (CPT or Model Growth), thereby leading to the observed scaling saturation. This phenomenon applies robustly across both CPT and model growth, extending findings previously confined to reinforcement learning ([2303.01486]).
> > - [2303.01486] also showed that layer normalization is most effective at mitigating plasticity loss. However, layer normalization is already used in our models, along with other regularization techniques such as weight decay and gradient clipping. **The most direct mitigation strategy is actually the one already highlighted in our practical implications** (Section 5.3) and discussion (Section 6): Our scaling law provides the quantitative tool to determine the optimal $D_1$ based on the available $D_2$ budget to avoid the negative effects. Future work can explore more techniques that explicitly target the sharpness of the loss landscape.

---

> > > ### Author Response · Authors · 2025-11-28
> > > **Reply (after significant update)**
> > >
> > > # General Comment
> > > We thank the reviewer for useful suggestions on our initial manuscript.
> > > We have performed a significant reorganization and revision of the manuscript to address issues raised by the reviewer.
> > >
> > > # Details
> > > > Limited Scale of Experiments
> > >
> > > In addition to our previous response on rigorous validation and budget consideration, we argue that the scale we studied is appropriate under reasonable budgets in **Appendix B.5**.
> > > This is because mapping the relationship between multiple scaling variables necessitates a multitude of controlled experiments on models of various sizes, in contrast to single, large-scale training runs (e.g., training one 7B model from scratch), which would provide fewer data points for fitting complex scaling laws.
> > >
> > > To strengthen our core claims under reasonable budgets, we further present more empirical results under various perspectives.
> > > - A new plot (**Figure 3**, left panel) which contains fitted scaling laws and marks the actual empirical observations explicitly. We explain the differences in the scaling behavior when comparing various functional forms with this plot.
> > > - Furthermore, we add a new plot (**Figure 5**, left panel). We show how the fitted scaling laws and empirical data points can be used as a guidance for mitigating the saturation effects.
> > > - We further tested the validity of our proposed scaling law at the limits, both analytically and with additional experiments run within a reasonable time and budget (new **Section 4.4 and Appendix E.1**)
> > >
> > > > Narrow Scope of Domain Shift
> > >
> > > - We include our previous response about the domains studied in **Appendix B.4**.
> > > - Furthermore, as shown in Table 1, we have tested with a variety of methods known to be scalable and/or best practices, and hence are confident of the generalizability of our claims. See **line 373**.
> > >
> > >  > Lack of Mechanistic Explanation
> > >
> > >  We have a dedicated sub-section (**section 4.1**) with content of our previous response to this matter.
> > >  Thanks for raising this issue, which we think has added significant quality to the paper!
> > >
> > > # Final Remarks
> > > We hope the above explanations are useful, and would be happy to provide more details.
> > > We believe we have significantly improved the paper due to your critical comments.
> > > We would appreciate it if the reviewer could reevaluate the work more favorably if these satisfactorily address the concerns.

---

### Official Review · Reviewer_gC4V · 2025-10-29

**Soundness:** 2
**Presentation:** 3
**Contribution:** 2
**Rating:** 4
**Confidence:** 4

**Summary:**

This paper proposes a new scaling law to describe bootstrapped pretraining—a unified term encompassing both continual pretraining (CPT) and model growth.
Unlike the classical power-law scaling, the proposed formulation introduces a logarithmic interaction term with respect to the first-stage dataset size D1:
L(D1,D2) = AD_1^(-alpha1) * D_2^(-alpha2+alpha3 * log(D_1)) + E
The key insight is that when the base model is over-trained (large D1), subsequent bootstrapped training suffers from saturation, yielding diminishing returns.
Extensive experiments confirm that this form achieves the lowest RMS fitting error among tested alternatives.
The paper further discusses practical implications for balancing bootstrapping versus training from scratch and for compute-optimal allocation.

**Strengths:**

1. The experiments convincingly demonstrate that including the log(D1)​ term improves fit quality across both continual pre-training (domain shift) and model-growth (no shift) settings.
2. The proposed scaling law reveals the impact of saturation, which contributes to more in-depth research on the best practice of multi-staged pre-training
3. The paper is clearly written, with well-structured sections and thorough comparisons against additive and hybrid baselines.

**Weaknesses:**

1. The mathematical intuition is not convincing enough: if the power laws are accepted (as condition 1 and condition 2), it should also be accepted that equation (2) should reduce to equation (6) when no distribution shift is involved. However, equation (2) does not model the impact of distribution shift (as the paper claims that it is also suitable for model growth with no distribution shift), so it seems not consistent with equation (6).

Perhaps the work will benefit from proving that (2) is better than (6) even when there is no distribution shift (perhaps with experimental results), and then it can be self-consistent. If this is the case, the top-down derivation part can be removed.

2. The practical implications are extrapolated from relatively small scales. Validation at larger model and data scales is necessary before treating these findings as practically predictive.

**Questions:**

1. As noted above, could you clarify the theoretical necessity of Eq. (2) over Eq. (6) when no distribution shift is present, and ideally provide experimental confirmation?

2. Regarding the second-stage training, is the learning-rate schedule re-warmed from scratch, or does it continue the remaining portion of the stage-1 schedule? (CPT can be vulnerable to learning rate setting for extremely large models (>30B), does eq.(2) still hold with less reasonable learning rate schedules?)

---

> ### Author Response · Authors · 2025-11-21
> **Reply**
>
> Thank you for the thoughtful review of our work. We are especially grateful for the reviewer's **positive evaluation of the convincing experiments, contribution to the best practice of multi-staged pre-training, well-structured sections and thorough comparisons against baselines** in our work.
> In the following, let us clarify questions and comments raised.
>
> ## Answers to questions
> > the theoretical necessity of Eq. (2) over Eq. (6)
>
> We appreciate the reviewer's careful scrutiny of the mathematical intuition.
> We agree that for continuous training (without dataset shift) of a fixed model size ($N$), the loss should ideally follow the simple law (eq. 6).
> However, in the context of our model growth experiments, **the system undergoes a fundamental "model shift"**: a base model of size $N$ is structurally modified into a larger model of size $N'$ before further training.
> The two-stage training process we are interested in is not a single, continuous optimization curve governed by a single parameter set.
> Therefore, the assumption that the result must **reduce to the simple additive form of Equation 6 is violated due to this "model shift", regardless of whether a data distribution shift occurs**.
>
> We have **empirically verified** that our proposed law (Equation 2) is a superior fit to the data, even in the absence of a data distribution shift: the RMS error we obtain when fitting with Eq. 6 for expansion and stacking methods are 0.032 and 0.025 respectively, an order of magnitude larger than those fitted with Eq. 2 in Table 1 (0.00279 and 0.00285).
>
> We acknowledge that the empirical superiority of Equation 2 strengthens the argument for a purely bottom-up, discovery-based validation (Section 4.1).
> However, we believe the top-down derivation (Section 3) should be retained because it provides the mathematical necessity for the functional form of Equation 2.
> It rigorously establishes that if one accepts the well-established power-law behaviors for both stages (empirically validated in Figures 1 and 3), the multiplicative form with the interaction term (Equation 2) is the **unique simplest candidate that satisfies these conditions and best fits the experimental observations**.
>
> *Please let us know if we have misinterpreted your question or there is anything unclear.*
>
> > is the learning-rate schedule re-warmed from scratch, or does it continue the remaining portion of the stage-1 schedule?
>
> **We studied both**: results presented in Section 3 have the learning rate re-warmed from zero. In Section 4, we study the case where the learning rate continues from the stable phase of the schedule/ For both cases, the optimizer states are reset (to emulate the realistic scenario where a base model's optimizer states are not available). We find that Eq. 2 **fits the best for both cases**.
>
> ## Weakness comment
> > limited scale
>
> We acknowledge the importance of large-scale studies; however, we respectfully argue that our current experimental scale is sufficient to validate our core findings and establish the proposed scaling laws.
> - We showed that **the scaling behavior is reliable and predictable**: We validated the joint scaling law across a range of model sizes, from 15M to 1B in our study. Figure 4 (Left) further showed empirically that **sub-0.5B models can reliably predict the loss behavior at the larger 1B scale**. This provides strong empirical justification for the robustness of our scaling laws. The scaling relationships derived from this validated space are then used to predict the behavior of much larger models (7B or more). This is the standard, accepted methodology in scaling law research, which allows researchers to inform the training of models far larger than those used in the original scaling study (e.g. Chinchilla paper).
> - **Budget constraints**: We note that our 1B parameter model already consumed a substantial portion of our available computational resources (estimated in Appendix B.4) as it requires extensive grid searches (we need to train many times on different models). Increasing by a single order of magnitude in both model size and dataset size would require multiple times the total GPU budget we have already spent, estimated to be more than tens of thousands of USD. Under the constraints of an academic research setting, such an expenditure is not currently feasible.
> - Previous influential work like the OpenAI scaling law paper also studied models only up to the sub-1B scale. Therefore, we believe it is reasonable to run multiple experiments (we have over **450 runs** in this paper) on model sizes up to this scale.
>
> We hope these addresses your concern. Please let us know if further clarification would be helpful.

---

> ### Author Response · Authors · 2025-11-28
> **Reply (after significant update)**
>
> # General Comment
> We are grateful for the reviewer's suggestions and candid assessment of our initial manuscript.
> The reviewer especially pointed out that *"practical implications are extrapolated from relatively small scales. Validation at larger model and data scales is necessary before treating these findings as practically predictive."*
>
> In response, we have performed a significant reorganization and revision of the manuscript to address this core issue.
>
> - We present the empirical evidence of the scaling law much more clearly and rigorously.
> - We reduce the emphasis on the more speculative results, primarily the original **Section 5.3** concerning the implications of extrapolated model size and training tokens, relocating them to the new **Appendix E.2 and E.3****.
> - We further tested the validity of our proposed scaling law at extreme limits, both analytically and with additional experiments run within a reasonable time and budget (new **Section 4.4 and Appendix E.1**)
> We detail our specific modifications with respect to the reviewer's feedback in the following.
>
> # Details
> > extrapolation without validation
> - We remove this part (particularly the old **section 5.3**), which are based on extrapolation, and hence are not empirically validated and rather confusing for readers, from the main text.
> - We instead produce a new figure (**Figure 3, left panel**), plotting fitted scaling laws and marking the actual empirical observations explicitly. We further validate and explain the differences in the scaling behavior when comparing various functional forms in this figure.
> - Furthermore, we add a new plot (**Figure 5, left panel**). We show how the fitted scaling laws and empirical data points can be used as a guidance for mitigating the saturation effects. This practical implication is *validatable*.
>
> The new **Section 4.2** contain such details.
>
> > model growth should have single continuous scaling law?
>
> We have addressed this in the previous response and we additionally included the discussion at **line 345** in the new manuscript.
>
> > top down derivation part not required
>
> Besides our previous response, we have also toned down the top-down derivation approach in the new manuscript.
> We now emphasize more on the empirical evidence of the bottom-up observation, and use the top-down derivation as a way to check if there is any other plausible functional forms satisfying natural constraints (new **Section 3.1**).
>
> > Validation at larger scales
>
> In addition to our previous response on rigorous validation and budget consideration, we argue that the scale we studied is appropriate with under reasonable budgets in **Appendix B.5**.
>
> This is because mapping the relationship between multiple scaling variables necessitates a multitude of controlled experiments on models of various sizes, in contrast to single, large-scale training runs (e.g., training one 7B model from scratch), which would provide fewer data points for fitting complex scaling laws.
>
> # Final Remarks
> With these adjustments, we have, under reasonable budgets, empirically stress tested our core claims thanks to the reviewer's feedback.
>
> We hope the above explanations are useful, and would be happy to provide more details.
> We would appreciate it if the reviewer could reevaluate the work more favorably if these satisfactorily address the concerns.

---

### Official Review · Reviewer_UxNn · 2025-10-31

**Soundness:** 3
**Presentation:** 2
**Contribution:** 3
**Rating:** 4
**Confidence:** 4

**Summary:**

The paper addresses the reuse of pretrained base models (bootstrapped pretraining) for improving a subsequent run on more data (continual pretraining, CPT) or more parameters (model growth). The authors build on the idea of *ossification*  under CPT, and discuss a general notion of *saturation* of a base model, such that any subsequent training does not provide a proportional marginal gain.
The power law loss relation is expanded with a combined effect on loss from tokens seen during the base model training *and* that seen in the second stage of bootstrapped pretraining.
Both analytically and empirically, the proposed functional forms for the scaling laws appear to match and accurately capture the core behaviour observed: overtraining a base model with more tokens leads to a *lesser* gain in loss in the second stage of training.
Thus, a practical guide emerges for when one should train a model from scratch versus start from a previously trained model (same size or smaller).

**Strengths:**

* Well motivated scope, and of relevant timing given the spurt in model growth and continual learning literature with scaling becoming more ubiquitous.

* The novel functional form to model loss given parameters and data from two training stages appear well established through empirical evidence.

* Training details and various design choices, along with the scaling fit procedures, are well reported.

* The finding can be reduced to a simple rule-of-thumb for practical deep learning under bootstrapped pretraining.

**Weaknesses:**

* Some of the notations used, especially with the overloading of $\alpha$ slow down the accurate parsing of the paper and thereby the understanding of parts of the analysis (refer to *Questions* below).

* Clarity: some of the claims and assumptions in the text are without citations, references, or more details (refer to *Questions* below).

* The equivalence of both continual pretraining (CPT) and model growth as bootstrapped pretraining procedures is not adequately explained. Despite the apparent adherence to the proposed functional form of the power-law loss, Table 2 and Figure 4 show the quality of fit being better for CPT.
  * This can veer into a longer discussion of the complete difference in both the CPT and Stacking second-stage training w.r.t. data and thus the learning scope. Therefore, it questions the interpretation of $L_{D_1}(D_2)$ as equivalent.

**Questions:**

Various questions and suggestions are enumerated below.

Please note, the rating will go up contingent on the following points, with higher weightage on: 3, 4, 6, 9.


1\.1. L47-49: Could the authors intuit here more, how and where was the adverse effect of overtraining on second-stage pretraining seen?

1\.2 L47-49: We are still observing a different scaling behaviour, as captured with the fitted parameters to the functional form (Eq. 2). Other than the power law requirement, what other `scaling behaviour` do we want to keep consistent (especially in the widely accepted *token-per-parameter* parlance)?

2\. L39,L109: The citation needs fixing.

3\.1. Figure 1: What is the model size here?

3\.2. Figure 1 (bottom left) and Figure 3 (right): Does this indicate that as long as the second stage training is *overtrained*, the training of the smaller model is irrelevant?

3\.3. How does this relate to the N, the parameters of the base model, in consideration?
* Could the main finding here be presented in relation to the Chinchilla, 20-tokens-per-parameter ratio for N and D?

4\. More generally, how does Table 7 or 8 compare to larger extrapolations of the scaling fits, beyond the validation data available? Especially when comparing the projected loss of more general functional forms, not capturing the interaction term of $D_1$ and $D_2$.
Alternatively, if possible, comparing projected loss (and a ($D_1$, $D_2$) recipe) from the fits obtained, to losses reported on training from scratch at similar (N, $D_2).

5\. L222-223: The Condition 2 for $L_{D_2}(D_1)$ should also be an equation for consistency. Adding subscripts to the $\alpha$s in both the equations should be useful.

6\. L241-244: The inline equality presented is hard to understand given $\alpha_1$ is not present in the RHS. Could the authors please clarify?

7\. The footnote 3 reveals a rather nice design choice in terms of a $\sqrt{2}$ scaling of the *aspect ratio* (depth/width) for model growth. This point could indeed be more explicit.

8\. L417: This comparison of training from scratch is only applicable for model growth and not CPT. Should the paragraph title reflect so and not bootstrapped pretraining generally?

9\. L436-446: Is there a limit to the performance of Stacking? That is, is there a slow down in convergence rates to the point a $2N$ model trained from Scratch achieves better loss with the same tokens seen?

10\. To extend 9 above, are the conclusions drawn, based on the empirical evidence collected, specific to the choice of methods used for model growth and CPT? Especially given that both growth and CPT are by now a varied body of literature. What do the authors think about this and therefore the claims made on *general bootstrapped pretraining*?

11\. Appendix B.3: What is the *standard practice* used here for batch size scaling? How are the warmups in Table 5 chosen, w.r.t. L805?

12\. Table 3: Is it normal for function-preservation on width-expansion under inconsistent head sizes (d_{model} / n_{head})?
* Could the authors clarify which method is being used in the experiments?

13\. Figure 10: The labels on the plot are confusing based on their alignments. Could that please be explained better in text or caption?

---

> ### Author Response · Authors · 2025-11-21
> **Reply 1**
>
> Thank you for the thoughtful and positive evaluation of our work. We are especially grateful for the reviewer's **appreciation of the well-motivated scope, novel functional form, well-reported experiments and practical guidance** of our work.
> In the following, let us clarify questions and comments raised.
>
> ## Answers to questions
> We first prioritize answering questions 3,4,6,9:
>
> > 3.1. model size
>
> The model size is 0.1B in Figure 1.
>
> > 3.2 the training of the smaller model is irrelevant if second stage is overtrained?
>
> We assume the question refers to the irrelevance of first-stage training ($D_1$) when the second stage is "overtrained" (i.e., when $D_2$ is very large).Roughly speaking, yes, the influence of $D_1$ diminishes as $D_2$ becomes very large.
>
>  However, the more critical finding is that the final loss will be smaller due to a higher $D_1$ because of the constant term $A_{D_1}$ (or $D_{1}^{-\alpha_{1}}$ in Equation 2), which pushes the entire learning curve lower.
>  More importantly, our scaling law, Equation 2 captures this dependency:
> - As $D_2 \to \infty$, the term $D_{2}^{-\alpha_{2}+\alpha_{3}log~D_{1}}$ approaches zero, making $L(D_1, D_2) \approx E$ (the irreducible loss) regardless of $D_1$.
> - However, in practice, for any finite, large $D_2$, the model with the largest $D_1$ still achieves the lowest final loss due to the scaling factor $A \propto D_{1}^{-\alpha_{1}}$.
>
> *Please let us know if we misinterpreted your question.*
>
> > 3.3. relation with N
>
> Our study extends the data-scaling analysis to include the model size ($N$) using the joint scaling law (Eq 7).
> - This joint law shows that the base loss term $BN^{-\beta}$ is additive to the data-dependent term3. This means the model size causes an offset on the final loss but does not affect the exponents ($\alpha_1, \alpha_2, \alpha_3$) that govern the power-law decay with respect to $D_1$ and $D_2$.
> - Our findings **directly inform the Chinchilla-style trade-off**. The conventional optimal token-per-parameter ratio is determined by the exponents of N and D. We show that for bootstrapped pretraining, the effective data exponent depends on $D_1$. This means the compute-optimal ratio changes depending on how long the base model was pre-trained, which is discussed in Section 5.3.
>
> > 4. larger extrapolations of the scaling fits
>
> We ask the reviewer to refer to Figure 10 (updated with an additional sub-figure) which provides a concrete illustration showing the results for a fixed base model size ($N=7$B).
> - Left: We compare loss-versus-token plots of from-scratch and bootstrapped pretraining at various base model training budgets (sunk costs).
>                 Each of the three colored lines (representing increasing sunk costs, $D_1$) crosses the dashed "Scratch" line at a different point.
>   - The crossover points demonstrate that the stacking model's **scaling efficiency diminishes with higher sunk cost** ($D_1$), eventually leading to the from-scratch approach becoming superior.
> - Right: We compare loss-versus-token plots of our proposed scaling law against other functional forms, with fixed sunk costs. We see that the predicted losses of the alternative functional forms increasingly diverge from our proposed scaling law. Furthermore, the predicted losses of other functional forms with different $D_1$ are **nearly parallel** across the range of $D_2$, whereas ours does not.
>   - This **lack of parallelism accurately captures the saturation effect**: as the base model is overtrained (higher $D_1$), the efficiency gain from additional $D_2$ tokens diminishes, leading to the lines converging at large $D_2$.
>
> > 6. L241-244: The inline equality
>
> We apologize for the typo  in the RHS of this equation. The term for the exponent of $D_1$ on the RHS should have included $\alpha_1$ :
>
> $$D_{1}^{-\alpha_{1}}D_{2}^{-\alpha_{2}+\alpha_{3}\log D_{1}}=D_{1}^{-\alpha_{1}+\alpha_{3}\log D_{2}}D_{2}^{-\alpha_{2}}$$
>
>  The intended purpose of the equation (despite the typo) was to illustrate the following property of the multiplicative form with an interaction term:
> - The multiplicative form with an interaction term is the most general form that satisfy both of our conditions, as the interaction term, $\alpha_{3}\log D_{1}$ satisfies Condition 1 (power law w.r.t $D_2$ for fixed $D_1$), while simultaneously, due to the symmetric property of the term (i.e., $D_{2}^{\alpha_{3}\log D_{1}}$ can be rewritten as $D_{1}^{\alpha_{3}\log D_{2}}$), satisfies Condition 2 (power law w.r.t. $D_1$ for fixed $D_2$).
> - Other simpler forms (additive and hybrid, Equations 3 and 4) fail to meet both power-law requirements simultaneously when the interaction term is introduced, thereby establishing the multiplicative form with this interaction term as the principled choice among the candidates.

---

> > ### Author Response · Authors · 2025-11-21
> > **Reply 2**
> >
> > > 9. Is there a limit to the performance of Stacking?
> >
> > The question of a performance limit for stacking is addressed by both Figure 4 (Right) and Figure 10, showing the effect holds across varying and fixed model sizes.
> > - Figure 4 (Right) presents the general limit by plotting the derived analytical and numerical solution for $D^*$ (the required tokens to train from scratch to catch up) across a range of varying base model sizes ($N$).
> >   - This figure shows that the relative advantage of model growth decreases as the sunk cost ($D_1$) increases and the base model size ($N$) increases.
> > - Figure 10 provides a concrete illustration of this general phenomenon by showing the results for a fixed base model size ($N=7$B). As explained in the answer to Q4, we interpret the results as follows:
> >  - Each of the three colored lines (representing increasing sunk costs, $D_1$) crosses the dashed "Scratch" line at a different point.
> >  - The crossover points demonstrate that the stacking model's scaling efficiency diminishes with higher sunk cost ($D_1$), eventually leading to the from-scratch approach becoming superior.
> >
> > > 1.1 intuition on how and where was the adverse effect of overtraining on second-stage pretraining
> >
> > The adverse effect of overtraining is seen as a diminished scaling efficiency (saturation).
> > - Intuitively, when a base model is overtrained, its highly optimized parameters make effective exploration after capacity enlargement for model growth (or exploration with new data distribution for CPT) difficult, causing the second stage to learn slower. Similar observation was made for the MoE case ([2409.02060]).
> >  - Our scaling law quantifies this by showing the $D_2$ exponent decreases logarithmically with the sunk cost $D_1$.
> >
> >  *Please let us know if we misinterpreted your question.*
> >
> > > 1.2 what other scaling behaviour do we want to keep consistent
> > - We impose minimal constraints: the functional form must maintain the power law relationship for both $D_1$ and $D_2$ independently.
> > - We do not assume further consistency; instead, our finding highlights the how the token-per-parameter ratio changes because the $D_2$ exponent depends on $D_1$ (section 5.3).
> >
> > > 2. The citation needs fixing
> >
> > Thanks. The reason seems to be due to a lack of entry in published year.
> >
> >
> > > 5. Adding subscripts
> >
> > Thanks for the suggestion. We will fix it in the upcoming version.
> >
> > > 7. The footnote 3 reveals a rather nice design choice
> >
> > Thanks!
> >
> > > 8. L417: This comparison of training from scratch for CPT
> >
> > Thanks for the comment. Although we could in principle train the model with, e.g., code data from scratch to compare it with CPT with code data, it is uncommon in practice. We will modify it in the upcoming version and show the uncommon practice only in the Appendix.
> >
> >
> > > 10.  conclusions drawn about bootstrapped pretraining
> >
> > We believe our conclusions on the scaling and saturation effects are generalizable across a variety of bootstrapped pretraining approaches, based on both the breadth of our experiments and established scaling law literature.
> > - While there might be edge cases that deviate (e.g., methods that are fundamentally unscalable or perform poorly), our study focuses on the practices considered scalable and representative of best practices ([2405.15319,2308.04014]). Therefore, we are confident that the scaling behavior and saturation effect identified here are **representative of general bootstrapped pretraining methods**.
> > - Scaling laws for LM are also known to be quite a general phenomenon insensitive to training/architecture details, as pioneered in the OpenAI scaling law paper.
> >
> > > 11. the standard practice used here for batch size scaling
> >
> > The standard practice we refer to is the practice of increasing batch size according to model size as in, e.g., Qwen2.5 paper.
> > As we round the batch size to numbers closest to the power of 2 for distributed training convenience, we do not perform precise tuning. The warmup steps are set approximately to be the same as the number of parameters following [2406.19146].
> >
> > > 12. Table 3: Is it normal for function-preservation on width-expansion under inconsistent head sizes (d_{model} / n_{head})?
> >
> > Thanks for the question and sorry for the lack of explanation in this part.
> > When increasing d, we also increase the number of head to keep the head sizes (d_{model} / n_{head}) consistent.
> >
> > > 13. Figure 10: The labels on the plot are confusing
> >
> > We will mention the following (as answered in Q.9):
> > - Each of the three colored lines (representing increasing sunk costs, $D_1$) crosses the dashed "Scratch" line at a different point.
> >  - The crossover points demonstrate that the stacking model's scaling efficiency diminishes with higher sunk cost ($D_1$), eventually leading to the From-Scratch approach becoming superior.

---

> > > ### Author Response · Authors · 2025-11-21
> > > **Reply 3**
> > >
> > > ## Weakness comment
> > >  > The equivalence of both continual pretraining (CPT) and model growth as bootstrapped pretraining procedures is not adequately explained
> > >
> > > - The quality of fit seems to be better for CPT is due to CPT and model growth being evaluated with **different test datasets**. CPT are evaluated with datasets with lower perplexity (or cross entropy), hence the absolute scale of error is smaller; the web dataset used to evaluate model growth has higher perplexity. The scale can be seen to be different in figure 7 (test loss at around 1.2 for code while 2.6-2.7 for web data).
> > > - indeed there are both different two-stage pretraining scenarios, with CPT dealing with dataset shift while model growth dealing with "model shift". We argue that under reasonable condition, our multiplicative scaling law with interaction fits the empirical observation the best, but do not compare the goodness-of-fit among these two different scenarios (which are **incomparable due to completely different settings**). However, the existence of an universal scaling relation is perhaps not too surprising, given that many different modalities of generative models satisfy a the same scaling law (with different parameters), e.g., [2010.14701].
> > >
> > > We hope these addresses your concern. Please let us know if further clarification would be helpful.

---

> > > > ### Comment · Reviewer_UxNn · 2025-11-22
> > > > **Reviewer Response to Author Rebuttal**
> > > >
> > > > Thanking the authors for their clarifications.
> > > >
> > > > B1\. L370-400: Formatting issue with whitespace and margins.
> > > >
> > > >
> > > > B2\. On point 3.2 response:
> > > >
> > > > > As $D_2 \textrightarrow \inf$, then $D_2^{\alpha_2 + \alpha_3 log D_1} \textrightarrow 0$, the term approaches zero
> > > >
> > > > This holds only when the exponent is lesser than 0, meaning, there is a upper bound on $D_1$ for this property to hold. More precisely, $D_1 < \exp^{\alpha_2 / \alpha_3}$.
> > > > Based on Equation 2, if $D_2 \textrightarrow \inf$ and $D_1 > \exp^{\alpha_2 / \alpha_3}$, then the first term in Equation 2 contributes extra loss over the irreducible loss, E.
> > > >
> > > > Would this then be contrary to the claims made by the author that empirically the model with the most $D_1$ would attain the best loss under $D_2 \textrightarrow \inf$.
> > > > If so, then the Equation 2 is an inadequate functional form.
> > > >
> > > >
> > > > B3\. On point 3.3 response:
> > > >
> > > > The general assumption for Equation 8 requires more clarification, and thus the role of $N$ and $D_1$.
> > > >
> > > > Why is the $D*$ assumption essential across both $D_1$ and $D_2$, and does that not affect the analytical expression derived for $D^*$?
> > > >
> > > > For the overall requirement to evaluate if training can be better than scratch, isn't the following more general requirements?
> > > >
> > > > $L_{N}^{\text{scratch}} (D_1 + D*) \ge L_N^{\text{grown}} (D_1, D*)$
> > > >
> > > > $L_{2N}^{\text{scratch}} (D*) \ge L_N^{\text{grown}} (D_1, D^*)$
> > > >
> > > > What I mean here is that from Section 5.3, the following claim,
> > > >
> > > > > We show that for bootstrapped pretraining, the effective data exponent depends on $D_1$
> > > >
> > > > does not follow, in my opinion.
> > > >
> > > >
> > > > B4\. On Figure 10: The updated plot is better, but still hard to read.
> > > >
> > > > (left) Is the red dashed "Scratch" line 14B model or 7B model; does the $D_2/D_{\text{scratch}}$ x-axis hold for this dashed line?; what exactly are the regions marked by $D_{\text{scratch}} < \approx > $ sunk cost?
> > > >
> > > > (right) Would more $D_2$ also lead to the orange and blue lines crossing each other, or would they always converge to similar losses? How is this related to large stacking of 3x or more?
> > > >
> > > > > This lack of parallelism accurately captures the saturation effect:
> > > >
> > > > Having read the paper, I understand what the authors mean here. However, we should use a different term other than `lack of parallelism` as it can be misleading or confusing.
> > > > I would suggest the authors consider phrasing this along the lines of: "converging fits at large $D_2$".
> > > >
> > > > Regarding plot correctness, I would advise the authors to mark the actual empirical observations explicitly, in order to denote what is a seen value and what is an interpolation in the plot.
> > > >
> > > >
> > > > B5\. On response to 9
> > > >
> > > > Figure 4 right predicts what (the y-axis)? That is, is it $D_2 - D_1$? If so, how does it only capture the relation of growing target model sizes and the base model tokens $D_1$?
> > > >
> > > > Given x-axis is the `base model size`, is it not natural that growing a model only slightly larger than the base model size will require almost the same extra tokens as the base model for a similar loss.
> > > >
> > > > I read this as: $L_{N+\delta}^{\text{growth}} (D_1, D_2)$, where $N+\delta << 2N$ (for large base model size as in the right of the Fig. 4 (right) x-axis).
> > > >
> > > > B6\. The authors could consider adding the following to the draft, along with references to the norm analysis,
> > > >
> > > > > Intuitively, when a base model is overtrained, its highly optimized parameters make effective exploration after capacity enlargement for model growth
> > > >
> > > > wrt the potential causes of the oversaturation effect.
> > > >
> > > >
> > > > ---
> > > >
> > > > The authors did a good job of clarifying much of the points raised in the original review and I thank them for it. However, for now, I would keep my score the same.
> > > >
> > > > Unfortunately, given that the saturation claim and the effect of $D_1$ on the role of $D_2$ in bootstrapped pretraining, *is* the core claim, I must admit that this requires further streamlining and clarity.
> > > >
> > > > For starters, improving Figure 4 and Figure 10 would go a long way.
> > > >
> > > > Clarifying Section 5.3 (which was referred to multiple time in the author rebuttal) better looks important too.
> > > >
> > > > The strong claim of the accuracy of the functional form as general might require more analysis at the limits (analytically perhaps), or pure empirical observations at even larger scales (at the cost of more compute so not a strict requirement).
> > > > Alternatively, toning down the claim and contribution while highlighting under a `Limitations` section where this *may not* work would be recommended.

---

> ### Author Response · Authors · 2025-11-28
> **Reply 1 (after significant update)**
>
> # General Comment
> We are extremely grateful for the reviewer's detailed suggestions and responses to our rebuttal.
> We here summarize what we did to improve our manuscript in response.
> As hinted by the reviewer, we realize that our manuscript mixes what we observe empirically with many extrapolative (unobserved) results (original section 5.3), making our core claim unclear and weak.
> - We significantly reorganize our manuscript to present the empirical evidence of the scaling law more clearly, including fitting plots with interpolative data points. We reduce the emphasis on more speculative results, i.e. implications of extrapolated model size and training tokens.
> - Also, our "top-down" approach to establish our claim might be too strong. We tone it down, and emphasize the top-down approach as a way to validate our claim compared with other plausible functional forms. We further test more rigorously the validity of the scaling law at the limits, both analytically, and with a few more experiments run within reasonable time and budget.
>
> We detail our modification with respect to feedback raised by the reviewer in the following.
>
> # Response to detailed points
> > B.2 the limits of the proposed functional form (our scaling law)
>
> We first would like to emphasize that we do not expect the scaling law to be true for all parameter regions (we have reorganized the manuscript to emphasize more the empirics, in part to tone down our top-down approach making wrong impression that the scaling law could describe the loss perfectly). Note also that previous influential papers like OpenAI and Chinchilla scaling law papers have similar description on the limitation of the scaling law.
>
> We next provide two arguments on the limit of scaling laws at large $D_1$.
>   - The threshold where the effective exponent becomes zero occurs at $D_1= e^{\alpha_2/\alpha_3}$. Based on our fitted coefficients, $\alpha_3$ is typically at least an order of magnitude smaller than $\alpha_2$. In fact, the threshold is more than $10^{17}$ with the fitted values in Table 7. Since the current largest dataset size (preprocessed appropriately) is in the order of trillions, we argue that reaching $10^{17}$ tokens is unlikely in realistic scenarios.
>   - We further argue that we can model the thresholding behavior deviating from the power-law ansatz. We can replace $D_1$ with a saturated form, $\hat{D_1}$, defined as $\hat{D_1} = \left(D_1^{-1} + D_{\rm max}^{-1} \right)^{-1}$, inspired by previous work on MoE scaling laws.
>
> **Large second-stage tokens.** We further study the limit where $D_2$ is large (wrt to model size, i.e., large token-to-parameter ratio), also answering partially **B4** on what happens after crossovers of lines with different $D_1$. We find that the losses at this limit can be described by our scaling laws too albeit slightly underestimated.
>
> These limitations are discussed in detail in the new **Section 4.4 and Appendix E.1**.
>
> > B.4 updated plot is better, but still hard to read
>
> We completely remove these plots which are based on extrapolation, and hence are not empirically validated and rather confusing for readers.
>
> We replace them with a new plot (**Figure 3, left panel**) using fitted scaling laws and marking the actual empirical observations explicitly. We explain the differences in the scaling behavior when comparing various functional forms, and use the description like *convergence* instead of *lack of parallelism* as suggested.
>
> Furthermore, we add a new plot (**Figure 5, left panel**), which performs scratch-versus-growth comparisons and is empirically verifiable (with fitted lines and interpolative data).
> We show how the fitted scaling laws (validated with empirical data points) can be used as a guidance for mitigating the saturation effects, strengthening our core claims. The new Section 4.2 contain such details.

---

> ### Author Response · Authors · 2025-11-28
> **Reply 2 (after significant update)**
>
> > B.3 and B.5: general assumption of equation 8
>
> We have moved this part to the **Appendix E.2**, as the region of interest presented in the plot is based on scaling law extrapolations (up to 1T parameters), making it more speculative and not directly essential to our core empirical scaling law claim.
>
> We would like to clarify the setup, which strictly follows the definition used in relevant prior work [2502.03009].
>
> It considers the specific case of stacking models with growth factor of 2, versus training from-scratch with a model with size equal to the *stacked* model.
> We do not change the growth factor. The x-axis always corresponds to the *base model size*. Moving on the x-axis by $\delta$ means that the base model size increases by $\delta$, and hence the  stacked model size increases by $2\delta$.
> The y-axis corresponds to $D_2$ for model growth or $D$ for from-scratch training.
>
> - The plot finds the minimum amount of additional tokens ( $D^\*$ ) required such that training a new, larger model from scratch ( $L_{2N}^{\text{scratch}}(D^\*)$ ) matches the performance of a model grown to that size after investing $D^\*$ tokens in the first stage ($D_1$) and $D^\*$ tokens in the second stage ($D_2$): $L_{2N}^{\text{scratch}}(D^{\*}) = L_{N}^{\text{grown}}(D_{1}=D^{\*}, D_{2}=D^{\*})$
>
> - **Interpretation:** The resulting curve delineates the boundary where the "sunk cost" does not longer make model growth superior to starting from scratch. This simplified 2D plot is meant by the original paper to *quickly* visualize the trade-off across token number and model size without simultaneously plotting three variables ($N$, $D_1$, $D_2$), instead of visualizing the more motivated but complicated requirements like $L_{2N}^{\text{scratch}} (D^\*) \ge L_N^{\text{growth}} (D_1, D^\*)$ as suggested.
>
> Regarding the reviewer's implied suggestion of studying the more well-motivated $L_{2N}^{\text{scratch}} (D^\*) \ge L_N^{\text{growth}} (D_1, D^\*)$, is now explicitly presented in the new **Figure 5 (left panel)**, using our empirical data, and serves the same practical guidance for practitioners.
>
> We hope this detailed clarification addresses any lingering confusion. Since this comparison is based on extrapolation and a fixed, specific ratio, we believe its placement in **Appendix E.2** is appropriate for the revised manuscript's focus on empirical findings.
>
> > B.6 Add norm analysis
>
> We added a dedicated section (**Section 4.1**) for this analysis. Thanks for the suggestion!
>
> # Final Remarks
> We hope the above explanations are useful, and would be happy to provide more details.
> We believe that we have significantly improved the paper due to your critical comments.
> We would appreciate it if the reviewer could reevaluate the work more favorably if these satisfactorily address the concerns.

---

### Author Response · Authors · 2025-11-28
**Summary of Updated Manuscript**

**Update after discussion phase ended earlier than expected:**

Since the discussion phase ended earlier than expected, it unfortunately prevents us from receiving response from the reviewers after we made significant rebuttal to the points raised by the reviewers. We hereby make further updates to address and clarify the valuable points and questions raised by the initial set of reviewers by editing this Comment.
 We hope you have immediate access to the core arguments we were unable to discuss further.

**Summary:**

We thank the reviewers for their constructive feedback.

As hinted by the reviewers, our initial manuscript, among other weaknesses, mixed empirical observations with extrapolative results (original Section 5.3), making our core claim less clear and weakened the overall presentation.
We have **significantly reorganized and revised the manuscript** to prioritize the clear presentation of our empirical evidence at scale, while reducing the description of extrapolated implications.

Let us walk through the key modifications and how they address the issues raised (major modifications are highlighted in blue in the revised manuscript):


**Section 2. Experiments**:
- Section 2.1 describes the experimental setup as in the original manuscript.
- We added Section 2.2. This new section clearly deduces our proposed scaling law from empirical observations, specifically highlighting **Observation 1 (power-law scaling in $D_2$ ) and Observation 2 (the logarithmic dependence of the exponent on $D_1$ )**.
This frames our core claim as purely **empirical** and derived from a **"bottom-up"** analysis, directly addressing concerns such as those raised by **reviewer UxNn** regarding the empirically lesssubstantiated nature of the old formulation.

**Section 3. Scaling Laws for Bootstrapped Pretraining**:
- Section 3.1 is essentially "formulating scaling laws" in the old manuscript. It is now explicitly motivated by **checking if any other plausible functional forms** can simultaneously satisfy the two fundamental conditions (power-law scaling with respect to both $D_1$ and $D_2$).
- Section 3.2 (combination of the old section 4: data scaling laws and section 5.1: variants of bootstrapped training) now directly **compares an extensive set of experiments** on a variety of bootstrapped pretraining variants (e.g., CPT with replay, CPT from stable phase, model growth with varying factors) for the above functional forms.
  - We additionally add a visualization of the fits of different functional forms, with empirical data points plotted (suggested by **reviewer UxNn**).
  - We further mention that single continuous scaling laws do not fit CPT well on different datasets and does not hold for model growth either, even when using the same dataset, due to the change in model capacity (suggested by **reviewer gC4V**).

**Section 4. A Closer Look At The Scaling Behavior**:
The old Section 5 on practical implications, which relied heavily on extrapolation, are rather speculative and have been mostly removed, leaving only one example in Appendix E.2. This ensures the primary claims remain grounded in empirical observation:
- Section 4.1 provides a mechanistic explanation (suggested by **reviewer 5caV**)
- Section 4.2 presents the practical, scaling-law-based strategy for choosing between bootstrapping and training from scratch , using the model growth scenario (and empirically verified 1B model) as a primary example. This rigorously validatable approach should address concerns on empirical validity by  **reviewer gC4V**, and reflect suggestion by **reviewer UxNn** on plotting scaling behavior (with actual data points)
- Section 4.3 formulates the joint scaling law with model size as in the original manuscript.
- Section 4.4 discuss limitations of the scaling laws, exploring regions where our laws do not hold (suggested by **reviewer UxNn**). Further studies on limitations are given in **Appendix E.1**.

**Additional content in Appendix**:
**Appendix B.5** argues that the scale we studied is appropriate under reasonable budgets in. This is because mapping the relationship between multiple scaling variables necessitates a multitude of controlled experiments on models of various sizes, in contrast to single, large-scale training runs (e.g., training one 7B model from scratch), which would provide fewer data points for fitting complex scaling laws.



Detailed descriptions of the modification tailored to reviewers' feedback are available in our new responses to each reviewer.

---

### Meta-Review · Area_Chair_E8BD · 2025-12-10

**Summary:**

This paper introduces scaling laws for "bootstrapped" language models, which are ways to use pre-trained models to get a good initialization for further pre-training. This includes regular continued pretraining, where the architecture stays the same, and model growth, where the model's size is increase. The goal of that paper is to estimate the loss on the new domain as a function of D1, the number of tokens used for the initial pre-training phase, and of D2, the number of tokens used in the second phase.

The law, in logarithmic form, can be written as
$$\log(L - E) = \log(A) -\alpha_1 l_1 - \alpha_2 l2 + \alpha_3 l_1 l_2,$$
where $E, A, \alpha_1, \alpha_2, \alpha_3> 0$ are parameters estimated from empirical runs, from a grid of D1, D2 values, and $l_1 = \log(D_1), l_2 = \log(D_2)$.

This formula is empirically motivated by fig.1, in which the authors fit one scaling law as a function of $D_2$ for a fixed pretraining budget $D_1$: $L(D2, D1) = E + A_{D_1}D_2^{-\alpha_{D_1}}$, where $A_{D_1}$ and $\alpha_{D_1}$ are estimated for each value of D1. the authors observe that the empirical $\alpha(D_1)$ are decreasing with $D_1$, which leads them to propose the scaling law above.

The goal of this paper is very timely, as it is critical to propose scaling laws for bootstrapped models. The analysis and insights of the authors are also very interesting and quite well motivated, as pointed by the reviewers. The fact that the paper covers both regular continued pre-training and model growth is a clear asset.

The authors did a major rewriting of the paper for the rebuttal, making the paper very clear now. The added section 4.4 is particularly welcome to partially address the counter-intuitive properties of the law.

The main concerns from the reviewers were:
- UxNn, gC4V, 5caV : the intuition behind the law is unclear, and it does not satisfy several intuitive properties.
- gC4V, 5caV: relatively small scale (up to 1B models trained for < 100B tokens), which makes it hard to know if the scaling laws extrapolate in the large (D1, D2) regime
- 5caV: only a few pre-training domains are considered.

I want to expand on the first point, as I believe it is the most severe weakness of the paper.
Indeed, the proposed scaling law formula predicts the following very counter intuitive behaviors that are not observed in practice:
- If $D_1$ or $D_2$ are large enough, taking the other D to infinity leads to a loss that goes to $+\infty$, while clearly it should go towards a minimum. In the rebuttal, the authors argue that this only happens for values of $D_1$ that are extremely high, but this is still a hint that this scaling law cannot explain well what happens in practice.
- More critically, and this was not pointed by any reviewer: "For any two pretraining budgets $D_1' < D_1''$, there is a value of $D_2$ large enough such that $L(D_1', D_2) < L(D_1'', D_2)$", i.e., pre-training becomes *counter-productive* for large enough $D_2$. For instance, take fig.3, left. The scaling law predicts that the curves must cross, and at a large enough $D_2$ budget, the blue curve (small pretraining budget) becomes better than the green one (large pretraining budget). This is a very counter-intuitive fact with, if true, deep conceptual consequences. Unfortunately, the paper offers **little empirical evidence that such a crossing ever occurs**. The authors mention this in Appendix E.1, with very small scale models. We see that in that experiment, the models all converge to the same when $D_2$ is large enough, which is an important discrepancy with the proposed scaling law. Therefore, the scaling laws is, at best, unvalidated, and at worst, misleading. This is the same story for fig. 1 and 5: a crossing is predicted by the scaling law, but not shown in practice.


I therefore recommend resubmitting the paper, with the following changes:
- **added experiments that clearly address this very counter-intuitive crossing property** This important property should be studied in more details, for larger models, and the scaling law fit in the large D2 regime should be investigated,
- **larger scale experiments** If the authors are bottlenecked by their compute budget, they could also use smaller models with much larger values of D/N as considered in the paper.

As a final remark, the authors mention L272 that we should recover a power law in $D1$ as $D2$ goes to 0 in the formula of $L(D1, D2)$, and indicate that this is the case for all the considered formulas. This is wrong: for all considered formulas, we have $L(D1, 0) = +\infty$ or, for the multiplicative law, $L(D1, 0) = 0$ when $D_1 > \exp(\alpha_3/\alpha_2)$.

**Reviewer Concerns:**

# Lack of scale

The authors explained in the rebuttal that they are bottleneck by their compute budget. The y convincingly argue that a well crafted scaling law study with < 1B models is valuable to the community. The main issue left with scale is that the authors never show the crossing happen in practice, while the scaling laws formulas predict it should happen, For instance, it would be nice to see what happens if the models in fig.5 are trained for 2x longer: do we really see a crossing?


# Intuition behind the scaling law

I think that this is the main weakness of the paper, which has not been addressed.

# Few pre-training domain

The authors argue that they study a very practical case. Experiments on more diverse domains would strengthen the paper, but in my view this is not critical.

**Reviewer Scores:**

I do not expect reviewers to change their score; the author rebuttal consisted of reframing / explaining better their reasoning and experiments, but there are little additional experiments to alleviate the main concerns of the reviewers.

---

### Decision · Program_Chairs · 2026-01-26

Reject